# Proteomic analysis of cell cycle progression in asynchronous cultures, including mitotic subphases, using PRIMMUS

Tony Ly[1,2]*, Arlene Whigham[3], Rosemary Clarke[3], Alejandro J Brenes-Murillo[1], Brett Estes[4,5], Diana Madhessian[6], Emma Lundberg[6], Patricia Wadsworth[4,5], Angus I Lamond[1]*

[1]Centre for Gene Regulation and Expression, School of Life Sciences, University of Dundee, Dundee, United Kingdom; [2]Wellcome Centre for Cell Biology, University of Edinburgh, Edinburgh, United Kingdom; [3]CAST Flow Cytometry Facility, School of Life Sciences, University of Dundee, Dundee, United Kingdom; [4]Department of Biology, University of Massachusetts, Massachusetts, United States; [5]Program in Molecular and Cellular Biology, University of Massachusetts, Massachusetts, United States; [6]Science for Life Laboratory, Royal Institute of Technology, Stockholm, Sweden

*For correspondence:
tly@dundee.ac.uk (TL);
a.i.lamond@dundee.ac.uk (AIL)

**Competing interests:** The authors declare that no competing interests exist.

**Abstract** The temporal regulation of protein abundance and post-translational modifications is a key feature of cell division. Recently, we analysed gene expression and protein abundance changes during interphase under minimally perturbed conditions (Ly et al., 2014, 2015). Here, we show that by using specific intracellular immunolabelling protocols, FACS separation of interphase and mitotic cells, including mitotic subphases, can be combined with proteomic analysis by mass spectrometry. Using this PRIMMUS (PRoteomic analysis of Intracellular iMMUnolabelled cell Subsets) approach, we now compare protein abundance and phosphorylation changes in interphase and mitotic fractions from asynchronously growing human cells. We identify a set of 115 phosphorylation sites increased during G2, termed 'early risers'. This set includes phosphorylation of S738 on TPX2, which we show is important for TPX2 function and mitotic progression. Further, we use PRIMMUS to provide the first a proteome-wide analysis of protein abundance remodeling between prophase, prometaphase and anaphase.
DOI: https://doi.org/10.7554/eLife.27574.001

## Introduction

The mitotic cell division cycle is composed of four major phases, that is, G1, S, G2 and M. The phases are defined by two major events during cell division: DNA replication (S phase) and mitosis (M phase), with intervening gap phases (G1 and G2). The cell cycle is driven by the expression of key proteins, called cyclins. Generally, cyclin expression and function is restricted to specific cell cycle phases, driving temporally ordered phosphorylation of key substrates by interacting with their kinase partners, the cyclin-dependent kinases (CDKs). Temporally regulated degradation of the cyclins ensures that progression through the cell cycle is unidirectional. For example, cyclin A expression increases during S-phase, reaching a maximum in mitosis. During prometaphase, cyclin A is targeted for degradation by the anaphase promoting complex/cyclosome (APC/C), a multiprotein E3 ubiquitin ligase, thus restricting cyclin A-driven phosphorylation to S, G2 and early M-phase.

Mitosis can be further resolved into subphases (i.e. prophase, prometaphase, metaphase, anaphase, telophase and cytokinesis), which are characterised by the widespread reorganisation of subcellular architecture. For example, in prophase, duplicated centrosomes separate to form the poles of the mitotic spindle. Centrosome separation is dependent on the activities of several kinases, including Cdk1 and Plk1 (*Smith et al., 2011*), and on the microtubule motor protein Eg5 (*Sawin et al., 1992*). Improperly timed centrosome separation can lead to chromosomal instability, as shown in cyclin B2-overexpressing MEFs (*Nam and van Deursen, 2014*). From prophase to prometaphase, nuclear envelope breakdown occurs and spindle assembly begins. During prometaphase, proper kinetochore microtubule attachments form, which depend on the interaction of each kinetochore on the sister chromatids forming stable, end on attachments to spindle microtubules. The recruitment of Eg5 to spindle microtubules is promoted by the microtubule-nucleating protein TPX2 (*Eckerdt et al., 2008*; *Ma et al., 2010*, *2011*). Expression of a TPX2 mutant lacking the Eg5-interaction domain leads to defects in spindle assembly and mitotic arrest (*Ma et al., 2011*).

Scheduled degradation of proteins is crucial for linking chromosome alignment and segregation (*King et al., 1996*) and is therefore an important regulatory mechanism for maintaining genome stability during cell division. Indeed, disruption of the scheduled degradation of key proteins, such as cyclins A and B, can lead to nuclear abnormalities, hyperplasia (*Bortner and Rosenberg, 1995*) and chromosomal instability (*Nam and van Deursen, 2014*).

A major challenge with the biochemical analysis of mitosis is that mitotic subphase durations are on the minute timescale (*Sullivan and Morgan, 2007*), with cell-to-cell variability in dwell times. This has hampered attempts to perform a comprehensive analysis of protein abundance changes during mitotic progression due to the difficulty in obtaining highly synchronous cell populations in the various intra-mitotic stages in sufficient numbers for proteome analysis. Additionally, emerging evidence suggests that methods used to either synchronise, or arrest cells in mitosis, may induce artefacts that are not observed during an unperturbed cell cycle (*Ly et al., 2015*). It has also been shown that long-term spindle assembly checkpoint (SAC) activation can lead to what is termed either mitotic 'exhaustion', or 'collapse', due to sustained degradation in the presence of an active checkpoint (*Balachandran et al., 2016*).

Technological advances in mass spectrometry (MS)-based proteomics have enabled the large-scale detection and quantitation of proteins, including measurement of properties such as absolute abundances and subcellular localisation (*Larance and Lamond, 2015*). However, proteome measurements on either cultured cells, or tissues, reflect average values across a cell population and detailed information on biochemical state heterogeneity is obscured. A powerful approach for separating cellular subpopulations involves immunostaining combined with Fluorescence Activated Cell Sorting (FACS). CyTOF is one example combining the single cell separation power of flow cytometry with accurate mass measurement and quantitation by mass spectrometry (*Bendall et al., 2011*). In CyTOF, immunological reagents are conjugated with heavy metal isotopes. Cellular material is then atomised for mass analysis of individual atoms, which report on the abundance of the immunological reagent binding to cells. We note that CyTOF, while significantly increasing the number of protein antigens detected as compared with a fluorescence-based assay, currently does not achieve the same high depth of proteome coverage, extending to thousands of proteins, that can now be obtained by large-scale, MS-based proteomic methods (*Hukelmann et al., 2016*; *Bekker-Jensen et al., 2017*; *Tyanova et al., 2016*; *Wilhelm et al., 2014*). Furthermore, apart from lower proteome coverage, CyTOF also has more limited ability to identify post-translational protein modifications and is a target-focused approach, in contrast with the unbiased analysis provided by MS-based proteomics.

Until now, studies combining FACS and MS-based proteomics have mostly involved isolating cell subpopulations based on labelling of cell surface antigens, hence avoiding cell permeabilisation and fixation (*Di Palma et al., 2011*), (*Bonardi et al., 2013*). Problems with sample bias and protein losses have been encountered in studies using MS-based proteomic methods when cells are permeabilised and fixed (*Toews et al., 2008*), as required for immunodetection of intracellular antigens. Indeed, to avoid intracellular immunostaining, recent work sought to identify mitosis-specific cell surface markers that can be used with live cells to isolate mitotic cell subpopulations in HeLa cells (*Özlü et al., 2015*). However, as far as we are aware, neither fluorescence abundance reporter systems, nor cell surface markers, have been reported to distinguish effectively between mitotic

subphases, for example prophase, prometaphase, metaphase and anaphase, suitable for discrimination and purification by FACS.

Previously, we reported a comprehensive proteomic dataset measuring protein and mRNA abundance variation across interphase (G1, S, and G2 and M) of the cell cycle (*Ly et al., 2014*). To prepare cell cycle enriched cell populations, we used centrifugal elutriation (*Banfalvi, 2011*), a method that minimises physiological perturbation to cells and thus avoids indirect effects modulating gene expression associated with arrest procedures (*Ly et al., 2015*). While elutriation was effective for isolating interphase subpopulations from human leukemia cells for proteome analysis, there were limitations in resolution. In particular, mitotic cells are poorly enriched relative to G2 cells. In contrast, G2 and mitotic cells can be efficiently distinguished using intracellular immunostaining and flow cytometry (*Jacobberger et al., 2008*; *Pozarowski and Darzynkiewicz, 2004*).

Building on our previous work analysing the cell cycle regulated interphase proteome (*Ly et al., 2014*, *2015*), we present here a workflow for performing *p*roteomics of *i*ntracellular *immu*nolabelled cell *s*ubsets (PRIMMUS). Using PRIMMUS, we perform a proteome-wide analysis of changes in protein abundance and phosphorylation during interphase. Further, we perform the first proteomic characterisation of distinct mitotic substages, with high enrichment efficiencies of prophase, prometaphase and anaphase in human NB4 cells. All of these proteomic data are freely available, both as raw MS files via the ProteomeXchange PRIDE repository (http://www.ebi.ac.uk/pride, PXD007787) and as quantified protein-level data, via the Encyclopedia of Proteome Dynamics (www.peptracker.com/epd/), a searchable, online database (*Larance et al., 2013*; *Brenes et al., 2017*).

## Results

### Optimisation of cell fixation and permeabilisation for proteome analysis

We identified three steps in intracellular immunostaining procedures that have the potential to significantly impact peptide identification by MS-based proteomics: (1) irreversibility and/or chemical modifications associated with cell fixation, (2) loss of soluble proteins during permeabalisation and (3) interference from antibody-derived peptides (*Figure 1A*). Therefore, a series of experiments were performed to compare alternative fixation and permeabilisation parameters with respect to these effects.

We chose to fix cells with formaldehyde (FA), which has been used extensively for other MS applications, such as protein-protein crosslinking (*Larance et al., 2016*) and crosslinked immunoprecipitations (*Mohammed et al., 2016*; *Klockenbusch and Kast, 2010*). FA forms reversible crosslinks that can be broken efficiently at high temperatures. However, prior work on model peptides shows that high FA concentrations can produce irreversible chemical modifications that compromise identification by MS (*Toews et al., 2008*), (*Sutherland et al., 2008*). FA concentrations and fixation times vary significantly between common immunostaining protocols (*Stadler et al., 2010*; *Stadler et al., 2013*). Therefore, we tested a range of FA concentrations in human myeloid leukemia NB4 cells, employing SDS-PAGE, immunoblotting and total protein gel stains to assay for crosslinking efficiency (*Figure 1B*). This identified 0.5% as the minimum concentration of FA that fixes cells and produces high-MW PAGE-impermeable crosslinked products. As shown by immunoblotting, crosslinking results in α tubulin migrating at increasingly higher MW bands in a FA-dependent manner, with no monomer remaining at 4% FA (*Figure 1—figure supplement 1A*). A similar FA-dependent shift is observed for histone H3 (*Figure 1—figure supplement 1B*). These data show that while 0.1% FA is sufficient to observe crosslinked proteins, 4% FA is required to crosslink most of the total protein pool. However, high FA concentrations reduce the efficiency of reverse-crosslinking, as discussed below.

To test the efficiency of reverse-crosslinking, FA-fixed lysates were heated at 95°C for 30 min, electrophoresed under reducing conditions and total protein visualised by SyproRuby staining (*Figure 1C*). The 17-kD band that was lost in a FA concentration-dependent manner (*Figure 1B*, arrow), was recovered upon crosslink reversal (*Figure 1C*, arrow). However, high MW bands, indicated by an asterisk, are still observed at 4% FA after heating. Quantitation of the summed intensity within the top third of the gel indicates a ~20% increase in intensity in 4% FA, compared with 0% FA, likely due to irreversibly crosslinked high MW protein-protein complexes. Consistent with the Sypro Ruby stain data, immunoblots show that the pools of crosslinked α-tubulin and histone H3

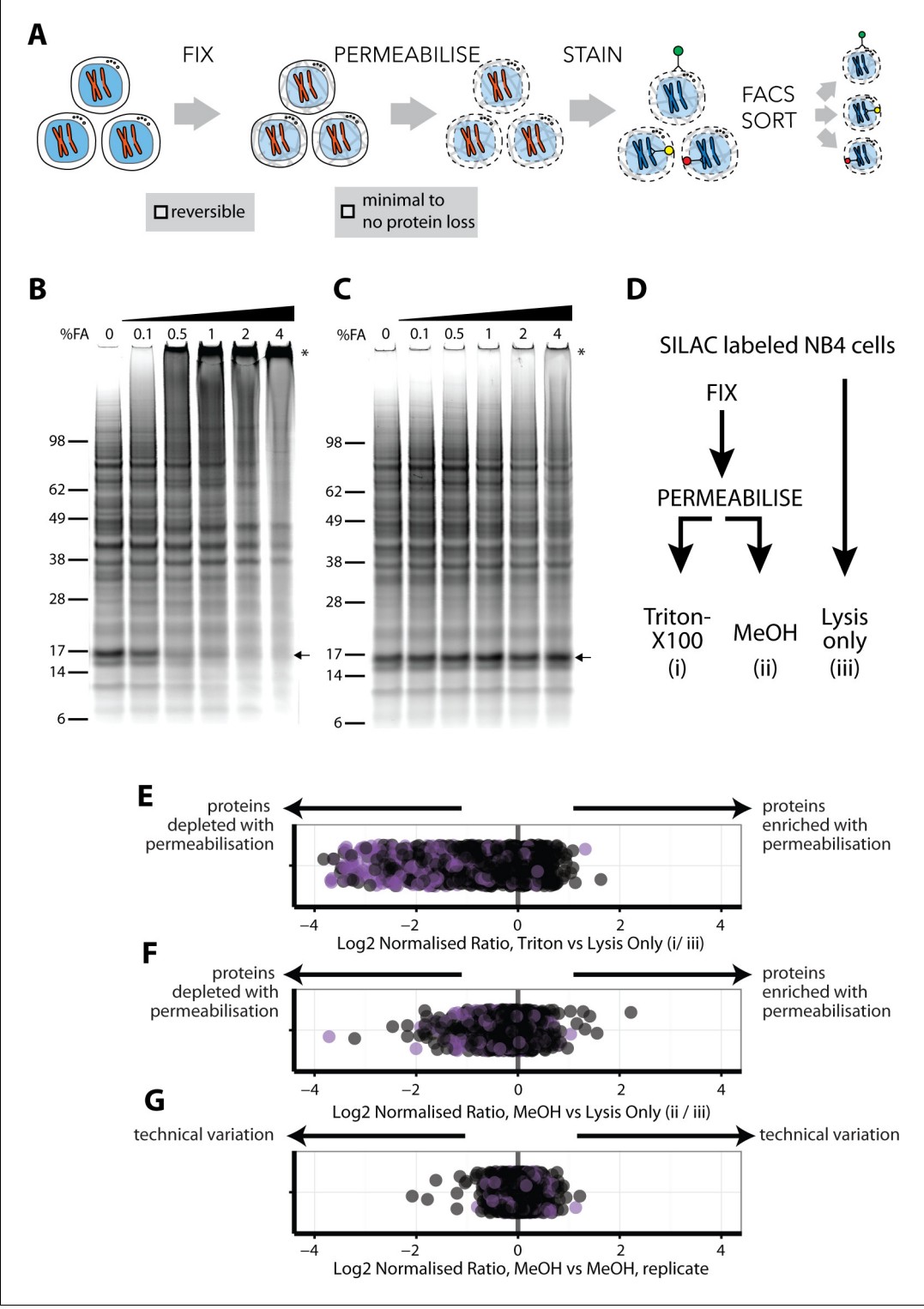

**Figure 1.** An optimised workflow for intracellular immunostaining, FACS, and MS-based proteomics. (**A**) An abbreviated schematic of the workflow for the Proteomics of Intracellluar IMMunolabelledcell Subsets (PRIMMUS) approach, highlighting specific steps for optimisation (fixation, permeabilisation). Lysates prepared from cells crosslinked with the indicated concentrations of formaldehyde (%v/v) in PBS (**B**) and then de-crosslinked with heating (**C**) were electrophoresed by SDS-PAGE and stained for protein using Sypro Ruby. (**D**). SILAC-labelled cells were either processed by fixation and permeabilisation, comparing 0.5% Triton X-100 (**i**) versus 90% methanol (**ii**), or with lysis only (**iii**). Cells were then mixed pairwise 1:1 and analysed by 'single shot' proteome workflows. The

*Figure 1 continued on next page*

*Figure 1 continued*

resulting SILAC ratios (e.g. H/L) are plotted as scatter plots for the pairwise comparisons, namely Triton X-100 vs. lysis only, methanol vs lysis only, and methanol versus methanol (technical replicate).

DOI: https://doi.org/10.7554/eLife.27574.002

The following figure supplements are available for figure 1:

**Figure supplement 1.** Immunoblot analysis of the effect of FA on the electrophoretic migration of individual proteins.

DOI: https://doi.org/10.7554/eLife.27574.003

**Figure supplement 2.** The effects of formaldehyde (FA) concentration on protein crosslinking and DNA staining.

DOI: https://doi.org/10.7554/eLife.27574.004

multimers, produced by treatment of cells with 4% FA, are heat stable (*Figure 1—figure supplement 1C and D*).

Protein-DNA crosslinks created by FA can affect the accuracy of measuring DNA content by flow cytometry and thus may decrease the resolution with which cell cycle phases can be separated. We measured the DNA content of cells fixed with the indicated FA concentrations, permeabilised with 70% ethanol and stained with either propidium iodide (PI, left), or with 4′,6-diamidino-2-phenylindole (DAPI, right), (*Figure 1—figure supplement 2*). Hoechst-33342 gave inferior results (data not shown). PI staining of cells treated with 0% FA produces low coefficient-of-variation (CV) values, for example 5.6% for G1, which increase in a FA dose-dependent manner. Peaks for 2N and 4N DNA content (i.e. G1 and G2 and M phases), coalesce at 1% and higher concentrations of FA. DAPI showed slightly higher starting CV values (7.3% for G1), but, unlike PI, was minimally affected by FA concentration. We conclude that, with these fixation conditions, 0.5% FA is the maximum for optimum use with PI, while 4% FA can be used with DAPI. In experiments where quantitation of DNA content by PI is not required to isolate a cell subpopulation of choice, then up to 2% FA can be used in combination with the intracellular staining protocol, with minimal protein loss (*Figure 1C*).

Quantitative comparisons of protein levels using stable isotope labelling by amino acids in cell culture (SILAC) (*Ong et al., 2002*), are internally controlled, thereby minimising the effect of technical variation after mixing on the accuracy of quantitation. We therefore used SILAC to evaluate how alternative permeabilisation reagents affect proteome recovery. Importantly, because SILAC quantitation is ratiometric and internally controlled, quantitation accuracy is not significantly dependent on potential protocol-dependent differences in overall proteome coverage achieved. For compatibility with both PI and DAPI staining and efficient crosslink reversal, cells were fixed with 0.5% FA. Three protocols were compared (*Figure 1D*): (i) cells were fixed and permeabilised with 0.5% Triton X-100, (ii) cells were fixed and permeabilised with 90% methanol and (iii) cells were lysed without fixation and permeabilisation. Cells were then mixed pairwise 1:1 by cell count and processed for 'single shot' MS analyses (*Supplementary file 1*).

The log$_2$-transformed normalised ratios for each of the ~1,235 proteins quantitated in all three conditions were compared (*Figure 1E–G*). In *Figure 1E*, the position of each point in the scatter plot relative to the x-axis represents the ratio between the abundance of a single protein measured in cells permeabilised with Triton X-100, (method i), and the same protein in cells processed by lysis only (method iii). Points are offset on the y-axis to minimise overplotting. Proteins detected equally in each method will show a ratio of 1, while irreversible FA-induced protein modifications and/or losses due to permeabilisation will cause lowered ratios. Proteins annotated with the subcellular localisation GO terms, 'membrane' and/or 'mitochondrial', are shown in purple.

Triton X-100 is widely used in protocols for cell permeabilisation when immunostaining intracellular antigens. However, we observe that many proteins (31%) are depleted in Triton-X100-treated cells (*Figure 1E*). Proteins showing lower ratios after Triton-X100 permeabilisation are enriched for 'mitochondrial' and/or 'membrane' GO terms (Fisher's exact test for enrichment: $p < 1 \times 10^{-25}$ for mitochondria and $p < 1 \times 10^{-20}$ for membrane GO annotations). In contrast, a comparison of methanol permeabilisation with direct lysis shows that most proteins (97%) vary <2 fold (*Figure 1F*). However, the levels of some proteins were reduced following methanol treatment. To investigate whether this was stochastic, we compared two technical replicates of methanol-permeabilised cells using SILAC (*Figure 1G*). The low variance between the replicates (s.d. = 0.15) indicates that the effects of methanol permeabilisation on protein extraction and measurement efficiencies are systematic and

reproducible. However, GO annotation analysis shows no significant enrichment for the proteins with decreased abundance after methanol permeabilisation (*Figure 1F*). The reason for the selective protein loss is therefore not completely clear but is unlikely to significantly bias downstream proteome analyses between samples that have been similarly methanol-treated.

In summary, we conclude that methanol is preferred over Triton-X100 as a permeabilisation reagent for downstream proteome analysis under these fixation conditions.

These data identify a protocol for immunostaining intracellular antigens that is compatible with efficient downstream MS-based peptide ID and quantitation and that minimises loss of protein identifications. While the fixation and permeabilisation steps slightly reduce the overall peptide signal (*Figure 1F*), these decreases are reproducible and can be accounted for in performing relative comparisons of protein abundance (*Figure 1G*). We term the resulting methodology using this optimised protocol, 'PRIMMUS' (*Pr*oteomics of *i*ntracellular *immu*nolabelled cell subsets).

## PRIMMUS analysis of protein accumulation across the cell division cycle

The PRIMMUS methodology is well-suited for transforming end-point immunostaining flow cytometry assays for cell cycle analysis into a preparative procedure for global proteome characterisation. For example, G1, S and G2 and M cell populations can be distinguished by DNA content alone using flow cytometry. As G2 and M phase cells have identical DNA content and similar size distributions, an additional parameter is required to separate these phases. H3S10ph, which accompanies mitotic chromatin condensation (*Hendzel et al., 1997*), is a specific marker for mitotic cells in many cell types and across many phyla (*Hans and Dimitrov, 2001*). H3S10ph staining is often used as a proxy for mitotic index, particularly in flow cytometry assays (*Juan et al., 1998*). The specificity of the anti-H3S10ph antibody for mitotic cells also aids an evaluation of the potential effect of antibody IgG peptides on peptide identification.

SILAC-labelled cells were fixed, permeabilised, immunostained and sorted into four subpopulations (i.e. G1, S, G2, and M), based on both DNA content and H3S10ph staining, then processed for MS analysis, as illustrated in *Figure 2A*. Four subpopulations are discernable in a representative psuedocolour scatter plot of the flow cytometric analysis, showing H3S10ph staining (y-axis), versus DNA content (x-axis) (*Figure 2B*). Cells were sorted into G1, S, G2 and M fractions using the sorting 'gates' indicated (*Figure 2B*, black boxes). The purities of G2 and M fractions were validated by co-immunostaining fractionated cells with anti-alpha tubulin antibodies and analysis by immunofluorescence (*Figure 2C*). This showed that none of the cells in the G2 fraction were mitotic and >96% of cells in the M fraction were mitotic, as evaluated by chromatin condensation and microtubule organisation (*Figure 2D*).

Relative protein abundances among the four sorted subpopulations of cells were determined using SILAC quantitation in single shot MS analyses (*Figure 2E*). Flow sorted 'heavy' cell populations were mixed with equal numbers of asynchronous 'light' cells, with the signal from the 'light' cells used as an internal standard to compare the four sorted populations (*Figure 2E*). Four biological replicates were performed.

In total, 32,066 peptides were identified from the four replicate experiments (raw data available at the ProteomeXchange Consortium partner PRIDE, identifier PXD007787). These peptide measurements enabled quantitation of 3,696 proteins, of which 3,162 have at least two supporting peptides per protein (*Supplementary file 2*). A comparable number of proteins were identified when the identical single shot MS analysis was performed on a standard complex peptide mixture prepared from directly lysed human NB4 cells (*Ly et al., 2014*). All cells in the asynchronous population are exposed to the H3S10ph antibody, yet only M-phase cells carry the antigen and are immunostained. Thus, the recovered M phase fractions will have IgG proteins that will be largely absent in the other fractions. Peptide ID and quantitation rates are similar across sorted populations (*Figure 2F*), suggesting that antibody-derived peptides present in the cell extracts from the M-phase fraction do not significantly hinder the MS-based peptide ID rate. Known FA-induced modifications, such as methylol and Schiff base intermediates (producing +30 and+12 mass shifts, respectively), were identified after adding these modifications to the search database. However, only a small number of peptides (<1%) were found to be modified (*Figure 2—figure supplement 1*).

Because the SILAC mixing was performed using equal numbers of cells from each population, rather than equal amounts of protein (as determined e.g. by mass or concentration), the ratios measured here reflect the variation in average protein abundance at the respective G1, S, G2 and M

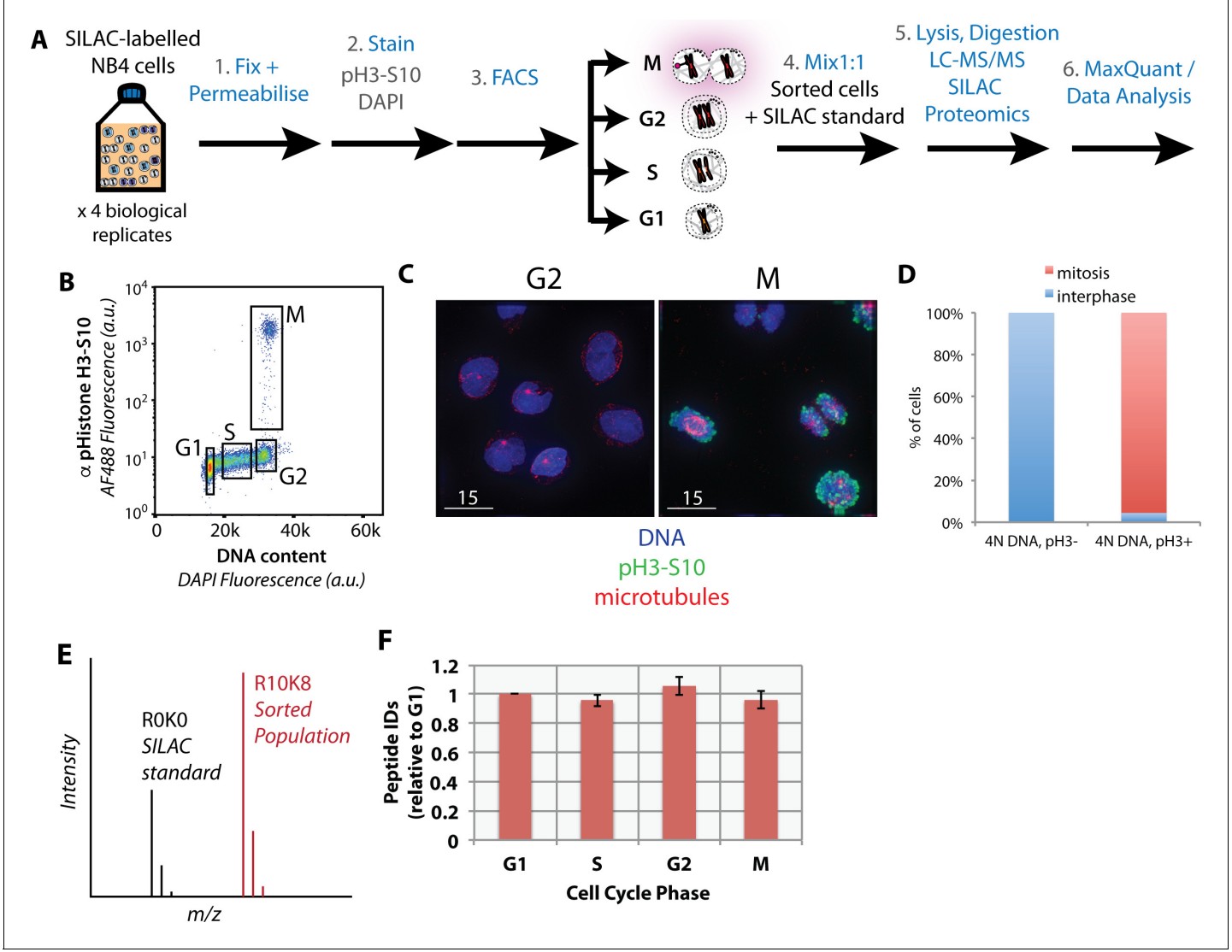

**Figure 2.** Purification of interphase and mitotic cells for PRIMMUS. Workflow for PRIMMUS of human leukemia cells into four cell cycle phase fractions (G1, S, G2, and M). Stained cells were sorted by FACS into four populations (G1, S, G2, and M) based on the gates shown on the psuedocolour plot in (B). The mitotic index of the M phase fraction was independently visualised by immunofluorescence microscopy and co-staining for microtubules (C) and quantitated (D). Fractions were then mixed by cell number 1:1 with an asynchronous SILAC-labelled standard and processed for 'single shot' LC-MS/MS-based proteomics. The resulting measured SILAC ratios compare protein abundances in the sorted fraction versus the asynchronous standard (E). The analysis was performed with replicates (n = 4). Comparison of peptide ID rates across the sorted fractions (F).

DOI: https://doi.org/10.7554/eLife.27574.005

The following figure supplements are available for figure 2:

**Figure supplement 1.** Sorting strategy.
DOI: https://doi.org/10.7554/eLife.27574.006
**Figure supplement 2.** Formaldehyde-induced modifications are generally low.
DOI: https://doi.org/10.7554/eLife.27574.007

phases during an unperturbed cell division cycle. *Figure 3A* shows a modified violin plot (here, called a 'neeps' plot) of $\log_2$ ratios versus sort population (G1, S, G2 and M). The width of each 'neep' is proportional to the number of proteins. Black lines mark quartiles and shading indicates interquartile ranges. Compared with the internal standard (asynchronous cells), G1 cells have a median ratio of ~0.8 and mitotic cells a median ratio of ~1.3. Thus, an average M phase cell contains ~1.6 x the number of protein molecules present in an average G1 cell. This value is less

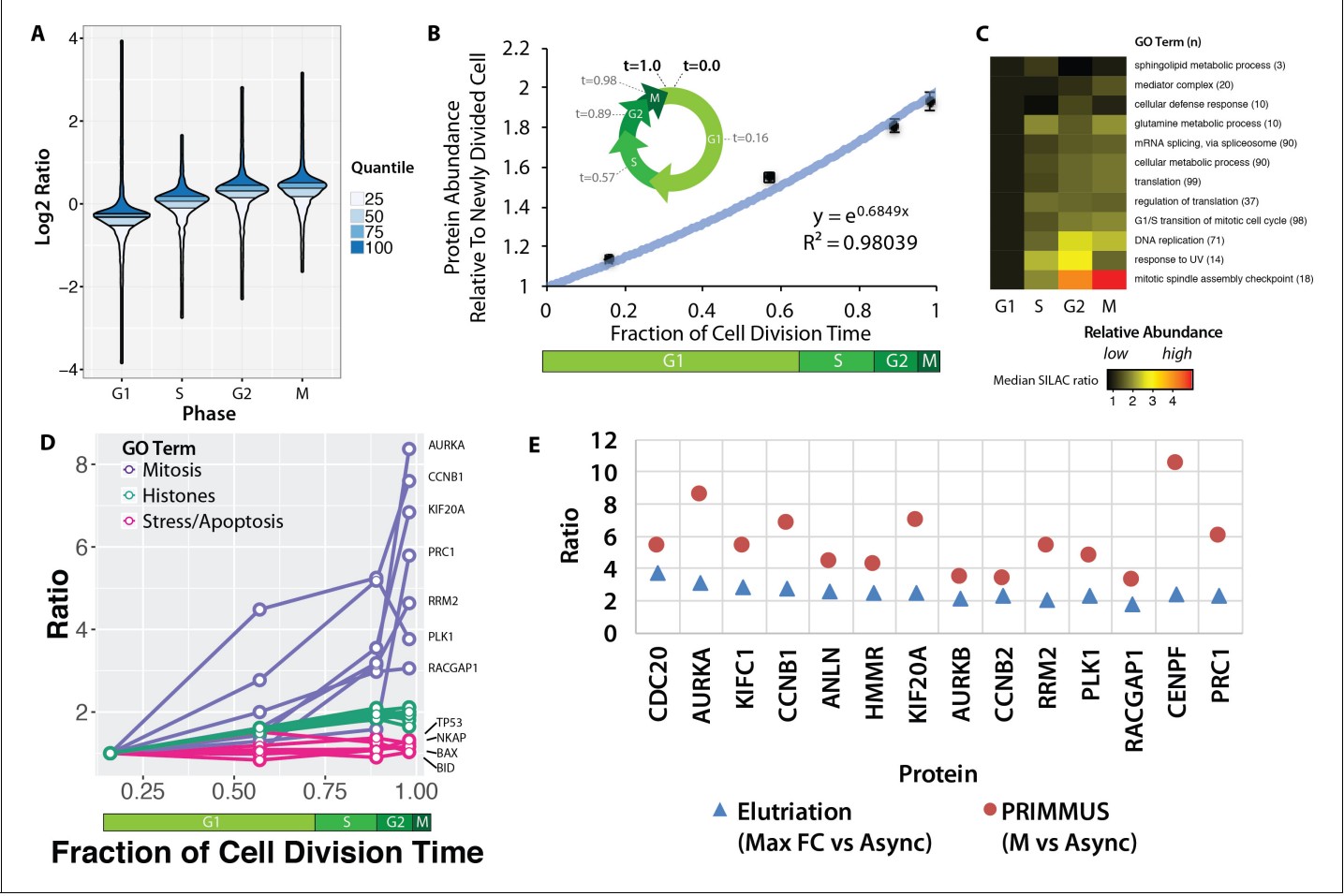

**Figure 3.** Proteomic measurement of protein accumulation across the cell division cycle. (**A**) A 'neeps' plot showing the distribution of $\log_2$ SILAC ratios measured in each of the G1, S, G2, and M subpopulations in one representative replicate. The width of each 'neep' is proportional to the density, that is the number of proteins. Quartiles are marked by black lines, and interquartile ranges are indicated by shading. B, inset) A schematic showing the cell division cycle and the average position during the cell division cycle for each phase collected, where a newly divided cell is defined as t = 0, and cell division (cytokinesis) is defined as t = 1. B, graph) Regression analysis was performed to produce a best-fit line in the form of an exponential growth model (i.e., $y = e^{mx}$). (**C**) Ratios of proteins belonging to each of the indicated GO terms were averaged (mean) and visualised using a heatmap. (**D**) A plot of ratios of individual proteins associated with mitosis, chromatin, and the stress response versus cell cycle stage. (**E**) A comparison of the ratio of G2 and M vs. asynchronous measured in the elutriation vs. PRIMMUS datasets.

DOI: https://doi.org/10.7554/eLife.27574.008

than 2 (the approximate ratio theoretically expected between a newly divided cell and a mitotic cell), because the 'average' G1 cell has already spent several hours in G1 phase, during which time new protein synthesis has commenced.

Using the average doubling time for NB4 cells (24 hr) and the frequency of cells in each cell cycle phase, as measured by flow cytometry (21%, 65%, 92% and 98%, for G1, S, G2, and M, respectively, from N = 5), we used ergodic analysis (*Pozarowski and Darzynkiewicz, 2004*; *Wheeler, 2015*; *Kafri et al., 2013*) to estimate the average time post-division for each subset of cells and express them as a fraction of the total cell division time, that is t = 0 (newly divided), t = 0.16 (G1), t = 0.57 (S), t = 0.89 (G2), t = 0.98 (M) and t = 1.0 (cell division). This representation of the relative cell cycle division time (*Figure 3B*, inset), allows protein abundance data to be plotted on a numerical time axis and provides a quantitative measurement of protein accumulation across the cell cycle.

Global changes in relative protein abundance at different cell cycle stages were estimated by taking the mean $\log_2$ SILAC measurement across all proteins. Mean protein measurements were calculated for each cell cycle phase, normalised to G1 and plotted as a function of cell division time, with

error bars indicating the standard error of the mean (*Figure 3B*). Bulk protein abundance accumulation across the cell cycle follows an exponential growth curve ($r^2 = 0.98$) (*Figure 3B*). Bulk protein abundance is measured to increase ~1.9 fold between a newly divided cell and a cell about to enter the next mitosis phase. These findings are consistent with results from experiments performed decades ago, using radioactive pulse-chase experiments to analyse bulk protein accumulation (*Scharff and Robbins, 1965*), (*Rønning et al., 1979*). However, using PRIMMUS, the relative protein accumulation across the cell cycle is measured here not on bulk protein alone, but rather on a per-protein basis for thousands of individual proteins.

We exploited the per-protein resolution of our data to examine whether different protein classes accumulate at different rates during the cell cycle. *Figure 3C* shows a heatmap summarising an analysis of the mean normalised ratio for proteins with GO annotations for major cellular processes, that is, transcription, mRNA splicing, translation, DNA replication and mitosis (*Figure 3C*). Proteins involved in DNA replication increase more on average during S phase than proteins involved in basal gene expression (mRNA splicing, cellular metabolism, translation). In contrast to most functional protein categories measured, sphingolipid metabolism and cellular defense response proteins remain relatively constant across the cell division cycle.

We show individual profiles for selected proteins in *Figure 3D*. Aurora kinase A, which regulates spindle assembly in mitosis (*Floyd et al., 2008*), increases in abundance by more than eight fold between G1 and M phase (*Figure 3D*). Core histones, however, show a profile that resembles bulk protein accumulation and approximately double in abundance across the cell cycle, consistent with the parallel doubling in DNA content. However, while bulk protein continues to accumulate between G2 and M (*Figure 3D*), histone abundance remains relatively flat, consistent with histone protein synthesis being markedly down-regulated once S phase is completed (*Figure 3D*). Interestingly, in contrast with bulk cellular proteins, we note that proteins involved in stress responses and/or apoptosis, such as p53, NKAP (a member of the NF-kappa B pathway) and the apoptotic regulators BAX and BID, remain relatively constant on a per-cell basis across these four cell subpopulations.

Using FACS, each cell cycle phase is enriched with higher purity as compared with centrifugal elutriation. We therefore analysed whether this results in the PRIMMUS method providing increased sensitivity for detecting relative changes in protein abundance between cell cycle phases. *Figure 3E* shows a comparison between datasets obtained using either PRIMMUS (red circles) or elutriation (blue triangles). These 14 proteins all peak in abundance in G2 and M relative to G1 phases, show the highest G2 and M vs. Async ratios in the elutriation dataset, and are annotated with 'cell cycle' gene ontology terms (i.e. CDC20, AURKA, KIFC1, CCNB1, ANLN, HMMR, KIF20A, AURKB, CCNB2, RRM2, PLK1, RACGAP1, CENPF and PRC1). All 14 proteins show higher ratios in the PRIMMUS dataset. We conclude that the higher enrichment purities for cell subsets obtained by FACS results in PRIMMUS providing a sensitivity advantage over centrifugal elutriation.

## PRIMMUS analysis of protein phosphorylation across interphase and mitosis

While formaldehyde-induced modifications are generally low (see above), the frequent reliance on single phosphopeptides for quantitation in phosphoproteomics means that phosphopeptide detection may be more challenging with fixed samples. We therefore assessed whether the PRIMMUS workflow was detrimental to phosphopeptide analysis. Lysates were generated from untreated cells and from cells processed using PRIMMUS. Proteins were precipitated, digested with LysC and Trypsin, and subjected to Ti:IMAC phospho-enrichment in technical triplicate, on different days. Enriched phosphopeptides were analysed using 2 hr LC-MS analyses detecting in total 6,587 phosphorylation sites and a mean of ~2,000 phosphorylation sites per individual analysis. *Figure 4—figure supplement 1* shows a comparison of phosphopeptide detection rates between control cells and cells processed for PRIMMUS analysis. No significant difference between the phosphopeptide identification rate was measured ($p > 0.05$, N = 3).

We next investigated protein phosphorylation changes during interphase and mitosis using PRIMMUS. Using the identical sort strategy as above, we separated asynchronous NB4 cells into G1, S, G2 and M phase fractions by FACS (*Figure 4A*). The experiment was performed in biological duplicate. The cell fractions were then processed for phosphopeptide analysis and TMT-based quantitation (8-plex, two biological replicates x 4 cell cycle phase fractions). Enriched phosphopeptides with no further peptide fractionation were detected and quantitated using a single shot LC-MS/MS

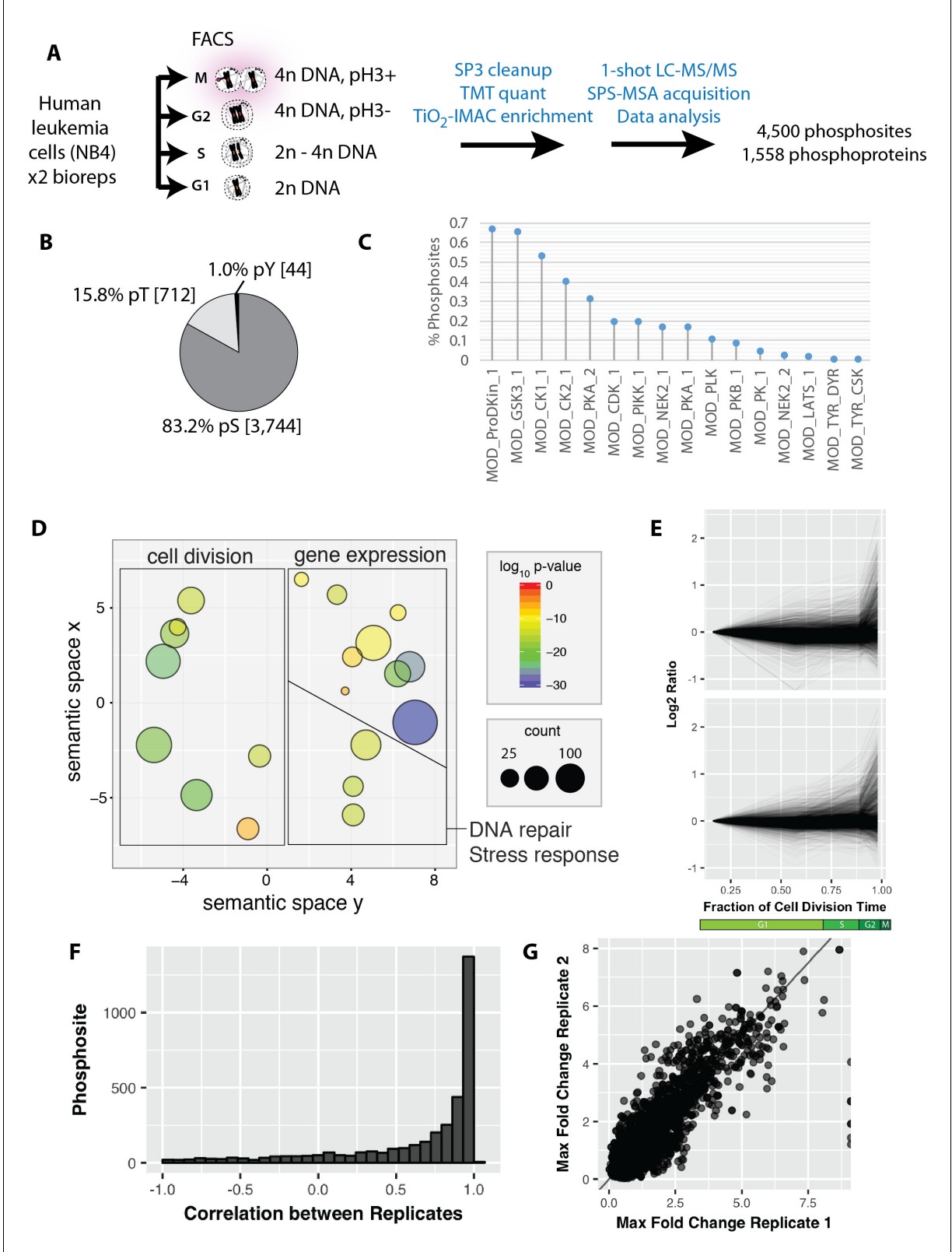

**Figure 4.** Reproducible analysis of phosphorylation changes across the cell division cycle.

DOI: https://doi.org/10.7554/eLife.27574.009

The following figure supplement is available for figure 4:

*Figure 4 continued on next page*

*Figure 4 continued*

**Figure supplement 1.** Comparison of phosphopeptide identifications between control and fixed, permeabilised samples shows no significant difference in identification rate.

DOI: https://doi.org/10.7554/eLife.27574.010

analysis. In total, 4,500 phosphorylation sites were identified on 1,558 proteins. Most phosphorylation sites were phosphoserines (83.2%), with smaller frequencies for phosphothreonines (15.8%) and phosphotyrosine (1.0%) (*Figure 4B*). Over 60% of the phosphorylation sites matched the proline-directed kinase motif (S/T followed by a proline (*Figure 4C*). CDKs and MAP kinases can phosphorylate the [S/T]P motif (as reviewed in [*Amanchy et al., 2007*]). The phosphoproteins detected are enriched in proteins with functions in cell division and DNA repair/stress response, in addition to 'housekeeping' proteins involved in gene expression (*Figure 4D*). The phosphorylation site abundance profiles for the duplicate measurements were similar (*Figure 4E*). As discussed in more detail below, many of the extreme differences in phosphorylation are observed in the mitotic fraction. Pearson correlations calculated for individual phosphorylation sites showed high correlation (*Figure 4F*). The maximum fold changes measured for individual phosphorylation sites were also highly correlated between the two replicates, showing that the quantitation was reproducible (*Figure 4G*).

The identified phosphorylation sites were then clustered using k-means into six groups (*Figure 5A*). Cluster one contains phosphorylation sites that rise significantly during mitosis. Interestingly, many of these sites also show an increase in the G2 fraction (*Figure 5A*, arrow). We note that a similar trend is observed when phosphorylation changes are normalised to protein abundance changes (*Figure 5—figure supplement 1*). Clusters that peak in mitosis (1, 2, 3 and 5) represent 34% of the phosphorylations quantitated (*Figure 5B*). Interestingly, this number is significantly smaller than reported previously in a study measuring phosphorylation dynamics in HeLa cells synchronised using nocodazole arrest and arrest-release protocols (*Olsen et al., 2010*). A comparison of the ratios measured for the same phosphorylation sites identified in the PRIMMUS dataset and in this previous analysis in synchronised HeLa cells shows significant differences. Phosphorylation sites upregulated in both datasets are on proteins enriched for the gene ontology annotations 'cell cycle' and 'mitosis'. We note that a similar enrichment for 'cell cycle' annotations was found for proteins whose phosphorylation sites specifically upregulated in this PRIMMUS dataset (*Figure 5C*, purple), which was performed using NB4 cells. In contrast, phosphorylation sites specifically upregulated in the previous HeLa cell arrest-release dataset (which are not changing in this PRIMMUS dataset), are instead on proteins showing significant enrichment for the gene ontology annotation 'RNA splicing function' and show no enrichment for either 'mitosis', or 'cell cycle' (*Figure 5C*, cyan). These differential enrichments in specific cellular functions for the proteins identified with changing levels of phosphorylation across the cell cycle suggest underlying physiological differences between the cells used in these respective studies (see also Discussion).

## Identification of a set of 'early rising' phosphorylation sites

While most of the significantly changing phosphorylation sites peak in mitosis, a subset, consisting of 115 sites, also show significantly increased phosphorylation in the G2 phase enriched fraction. We have termed these 115 phosphorylation sites, 'early risers'. The high enrichment efficiency of FACS (c.f. *Figure 2C*) renders it unlikely that the increased G2 ratios measured originate from contamination from H3S10ph-positive mitotic cells in the G2-enriched fraction. Indeed, further analysis shows that early risers share additional functional similarities. Thus, early rising phosphorylation sites are situated on proteins highly enriched in nuclear, nuclear envelope and chromatin localisations using identified phosphoproteins as 'background'(*Figure 6A*).

A STRING network analysis (*Figure 6B*) identifies several functional categories of early rising proteins, including DNA replication, cytoskeleton remodelers and spindle/kinetochore components, chromatin factors and remodelers, nuclear envelope proteins, transcription factors, nucleolar proteins and mRNA capping proteins. Proteins with the highest ranked ratios are either involved in DNA replication, or associated with the nuclear envelope (*Supplementary file 2*). Motif analysis (*Chou and Schwartz, 2002*) shows that early risers are enriched in the optimal CDK consensus motif of a serine/threonine followed by a proline and a C-terminal basic residue (*Figure 6C*). Consistent

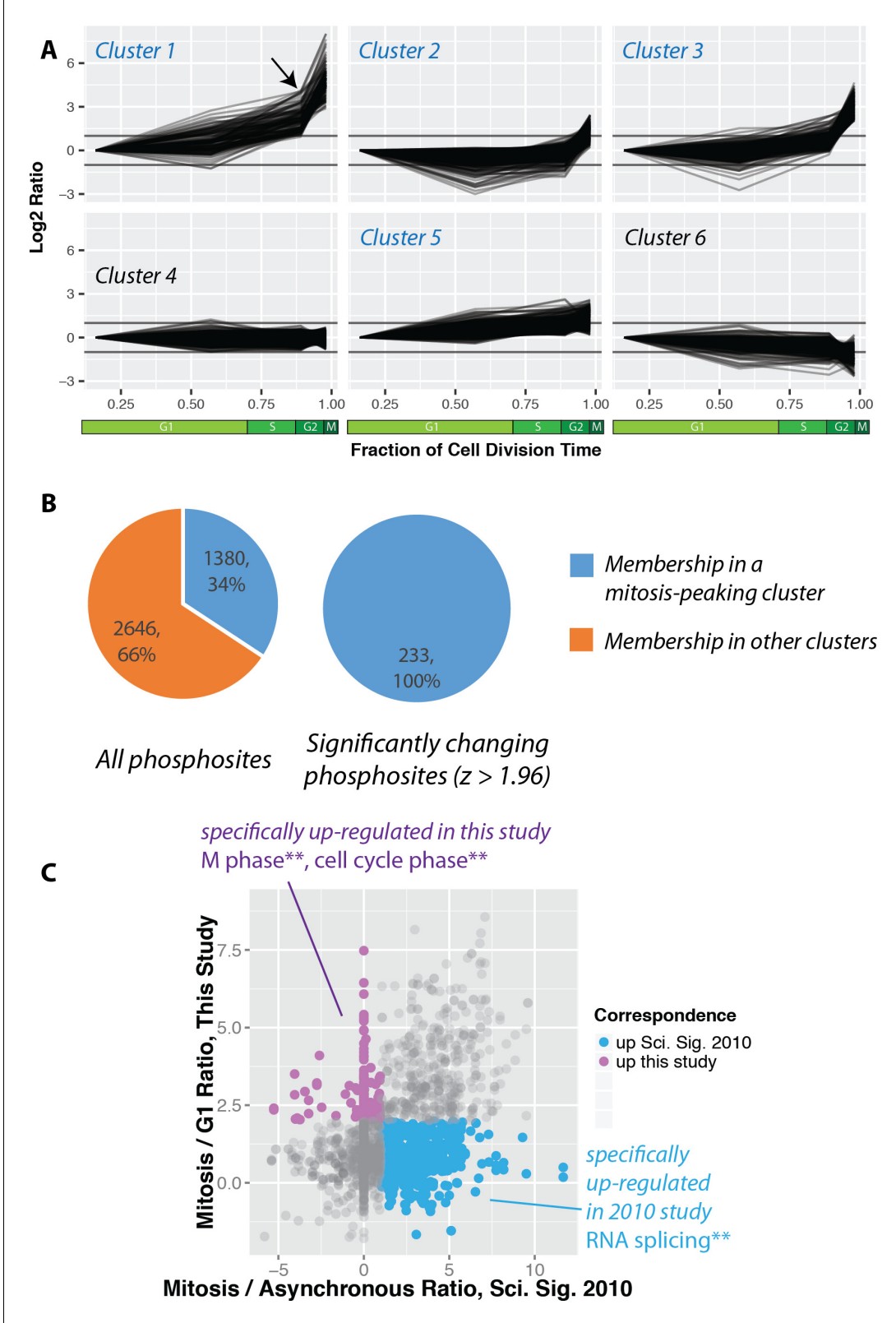

**Figure 5.** Increased global mitotic phosphorylation dominated by a subset of highly phosphorylated proteins. (**A**) K-means clustering of the phosphorylation profiles. (**B**) Distribution of mitosis-peaking phosphorylation sites, either in the entire dataset (left), or significantly changing phosphorylation sites (right). (**C**) A comparison of phosphorylation site ratios measured in this dataset and a previous analysis of mitotic phosphorylation in human cells.

*Figure 5 continued on next page*

*Figure 5 continued*

DOI: https://doi.org/10.7554/eLife.27574.011

The following figure supplement is available for figure 5:

**Figure supplement 1.** K-means clustering of the phosphorylation profiles normalised to total protein abundance.
DOI: https://doi.org/10.7554/eLife.27574.012

with a high CDK phosphorylation propensity, early risers on average have higher phosphorylation ratios during M-phase compared to 'late risers' (*Figure 6D*), that is late risers being defined as phosphorylation sites that peak in mitosis and do not increase in G2.

One of the early risers identified was on the protein TPX2. Two TPX2 phosphorylation sites are quantitated in this dataset, that is, S185 and S738. Both sites have surrounding sequences matching the consensus CDK phosphorylation motif (SPEK and SPK, respectively) and show increased phosphorylation in the G2-phase enriched fraction, compared with the total levels of unmodified TPX2 protein (*Ly et al., 2014*). However, TPX2 S738 is an early riser site that is phosphorylated to a greater extent in the G2 phase fraction and this difference is further increased in the M-phase fraction. We therefore decided to explore whether there was any functional relevance for phosphorylation of TPX2 at S738 in mammalian cells by analysis of phosphodefective and phosphomimetic mutants.

## Expression of S738A TPX2 mutant fails to rescue TPX2-depleted cells

To assess the potential function of TPX2 phosphorylation at serine 738, the mitotic phenotype of porcine LLC-Pk1 cell lines expressing TPX2 or TPX2 mutants was analyzed. A mouse bacterial

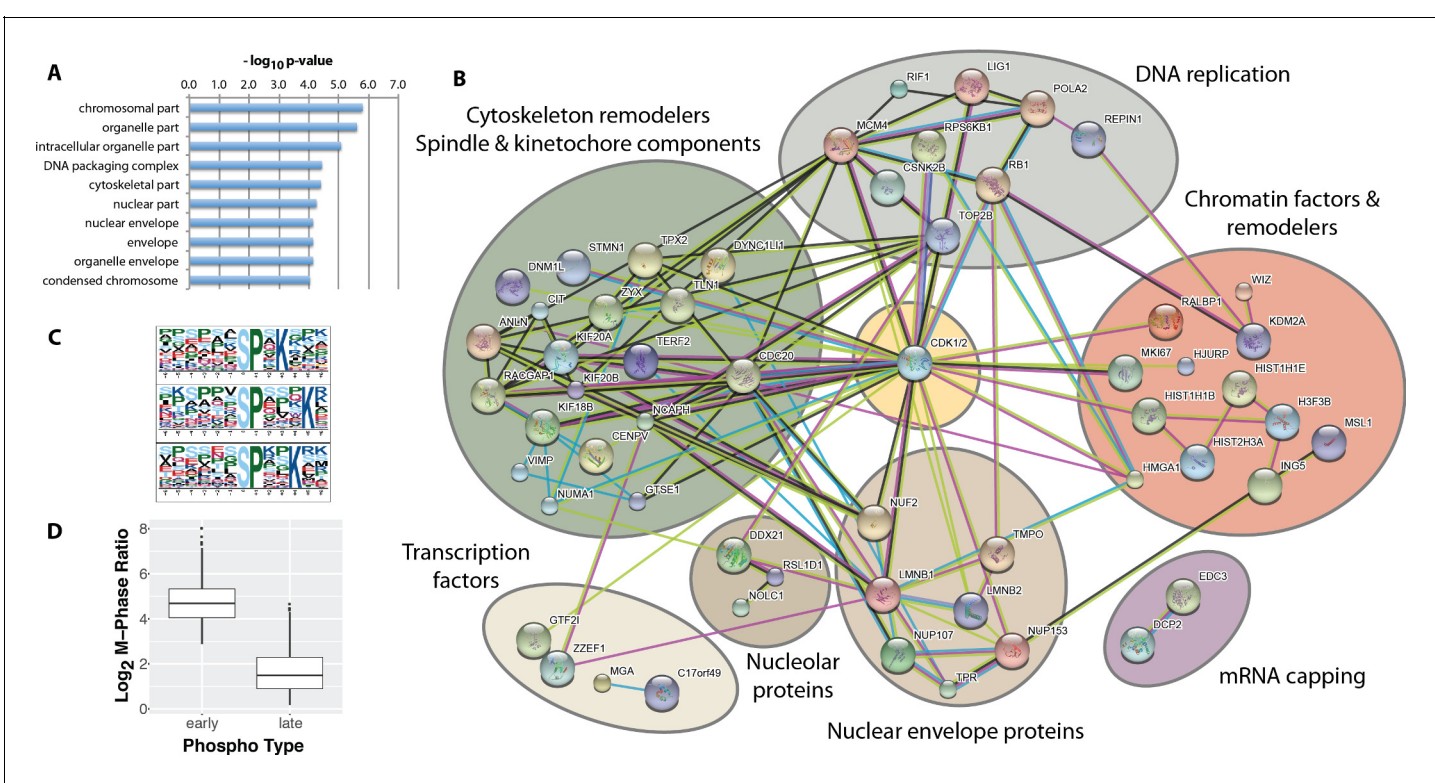

**Figure 6.** Identification of 'early risers', a subset of mitotic phosphorylations that begin increasing in G2 phase. (**A**) Gene ontology enrichment analysis of early rising phosphorylation sites. (**B**) A STRING network analysis of early rising phosphoproteins. Nodes with one or more connections are shown. (**C**) Enriched sequence motifs among early rising phosphorylation sites (Motif-X). (**D**) Comparing the M-phase ratio between 'early rising' and 'late rising' phosphorylation sites. Error bars show s.e.m.
DOI: https://doi.org/10.7554/eLife.27574.013

artificial chromosome (BAC) expressing wild type TPX2 with a GFP tag was mutated to generate either a phosphodefective TPX2-738A-GFP mutant, or a phosphomimetic TPX2-738D-GFP mutant. Cell lines expressing each of the constructs were generated. To examine spindle phenotypes, endogenous porcine TPX2 was depleted using siRNA (*Ma et al., 2011*) and spindle morphology was scored in either live, or fixed cells (*Figure 7*).

Consistent with prior work, depletion of TPX2 from parental cells resulted in a collapsed spindle phenotype characterised by two large asters with a short intervening spindle (*Gruss et al., 2002*). Spindles in cells treated with a non-specific siRNA were nearly all bipolar (*Figure 7*). Expression of either of the mutant forms of TPX2 resulted in defects in spindle morphology including monopolar, multipolar, bent and misshapen spindles (collectively referred to as 'other', *Figure 7*) as well as collapsed spindles. The results show a decrease in the percentage of bipolar spindles and increase in defective spindles for both mutations; neither phenotype was as severe as cells treated with siRNA targeting TPX2 alone (*Figure 7*).

To further examine the consequences of mutation at serine 738 in more detail, time lapse imaging of cells expressing either wild-type, or mutant TPX2, and treated with siRNA targeting porcine TPX2, was performed; only cells that completed mitosis during the imaging period were scored. Although the mitotic duration of the phosphodefective 738A mutant was not different from that of cells expressing the wild-type protein, a mitotic delay was observed in cells expressing the phosphomimetic version of TPX2 consistent with a role for the site in mitotic progression (*Figure 7*).

Because the TPX2-738 phosphorylation site was classified as an early riser, a defect early in mitosis might be expected. TPX2 is nuclear throughout interphase, although a small fraction of the protein can be detected outside the nucleus, at the centrosome, in prophase (*Ma et al., 2011*; *Vos et al., 2008*); however, the contribution of TPX2 to the events of early mitosis is not yet established. Defects in mitotic progression prior to NEBD would be not be detected in our time-lapse imaging. However, the observed spindle defects are consistent with deficiencies in spindle formation and a contribution of the TPX2-738 site to mitosis in live cells.

## PRIMMUS analysis of protein abundance variation during mitosis

We next used the PRIMMUS method to perform a proteomic analysis of protein abundance variation across four temporally distinct stages of mitosis, reflecting prophase, prometaphase (1 and 2, see below) and anaphase. We aimed to screen for proteins that show abundance patterns resembling cyclins A and B, which could reveal novel targets whose degradation is also regulated during mitosis. We chose H3S28ph and cyclin A (CycA), as two markers with which to distinguish different mitotic subphases. Like H3S10ph, the H3S28ph signal is associated with chromatin condensation (as reviewed in [*Hans and Dimitrov, 2001*]). During mitosis, cells undergo reversible condensation of chromatin, with highest levels of compaction observed during prometaphase and metaphase (*Hans and Dimitrov, 2001*). Thus, cells showing the highest levels of H3S28ph signal (H3S28ph-high), represent prometaphase and metaphase cells, while cells showing intermediate levels of H3S28ph signal (H3S28ph-mid), are in early (prophase) and late (anaphase and telophase), stages of mitosis, respectively. Meanwhile, CycA is targeted for degradation by the APC/C during prometaphase in a SAC-independent manner (*den Elzen and Pines, 2001*). Thus, comparing either the presence, or absence, of CycA provides a means for distinguishing between 'early' (prometaphase and before) vs. 'late' (prometaphase and after) mitotic cells, respectively. Consistent with this, flow cytometry analysis of cells co-immunostained for H3S28ph and CycA show four subpopulations (labelled P1 – P4), which are H3S28ph positive (*Figure 8A*).

The four subpopulations described above were isolated by FACS and analysed by immunofluorescence microscopy (representative images are shown in *Figure 8A*, right). The frequency of interphase, prophase, prometaphase, anaphase and telophase cells were each measured, with at least 100 cells counted for each subpopulation (*Figure 8B*). This confirms that high enrichment efficiencies were obtained for prophase, prometaphase and anaphase, respectively. Telophase and cytokinesis cells were not observed in these subpopulations. We note that the gating strategy employed, which removes potential doublets from the analysis, biases against these cells (unpublished observations). *Figure 8C* shows a representative image of the P4 subpopulation, showing high enrichment of anaphase cells. Based on these high enrichment efficiencies, we have relabelled these populations according to the major enriched phase represented, that is, prophase (Pro), prometaphase 1 (PM1), prometaphase 2 (PM2) and anaphase (Ana), respectively.

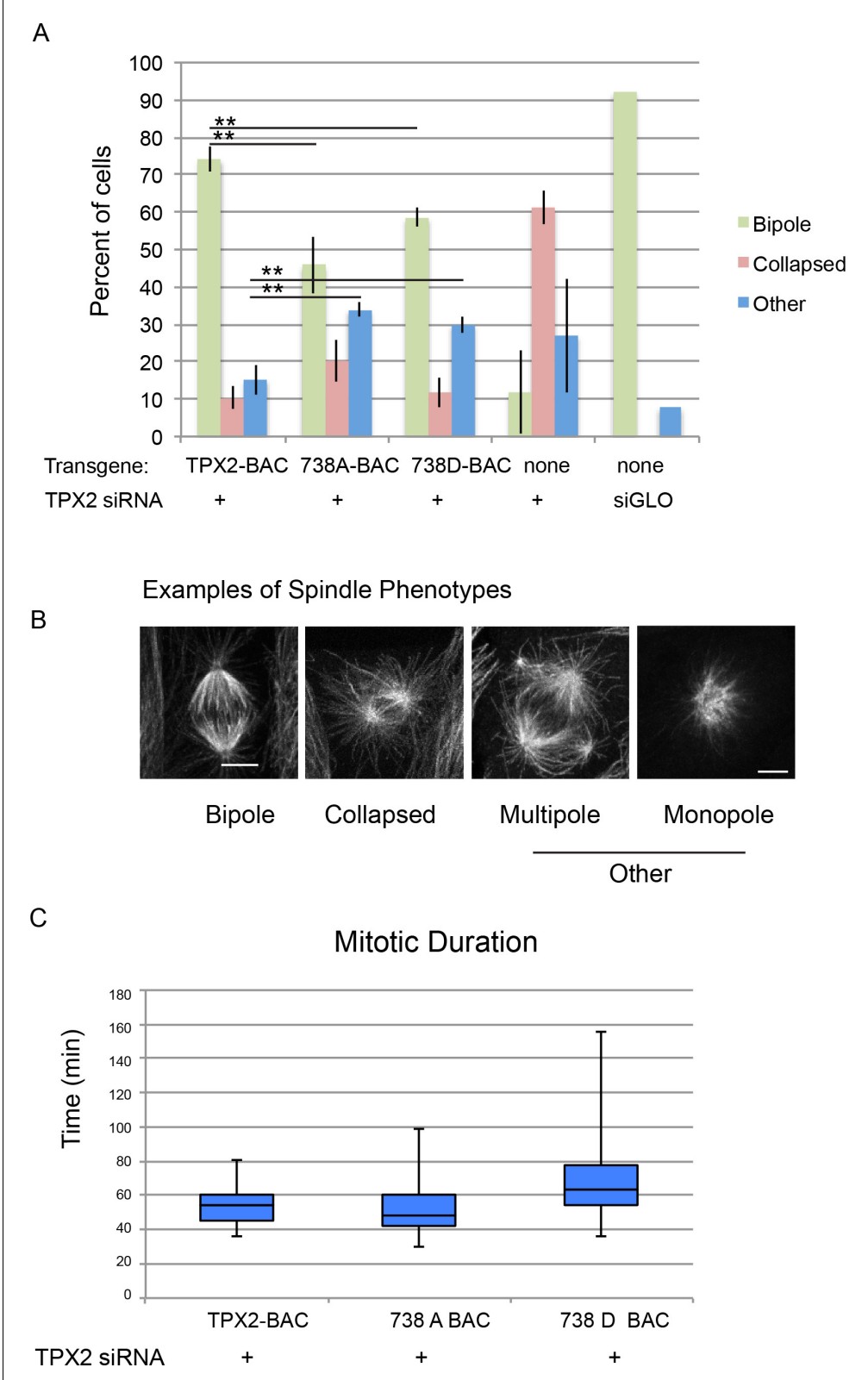

**Figure 7.** Mitotic defects in cells expressing phosphodefective and phosphomimetic mutants of TPX2-S738. (**A**) Bar graph showing percentages of mitotic spindles with the indicated phenotypes. Collapsed spindles have large asters and short intervening spindle; spindles classified as other include monopolar, multipolar, bent and
*Figure 7 continued on next page*

*Figure 7 continued*
misshapen spindles. (**B**) Representative examples of spindle phenotypes. Bar = 5 microns. (**C**) Box and whisker plot of mitotic duration, defined as nuclear envelope breakdown to anaphase onset.
DOI: https://doi.org/10.7554/eLife.27574.014

The FACS protocol was repeated, using three separate asynchronous cell populations cultured and harvested on different days. Two of these populations were metabolically-labelled with stable isotope labelled amino acids for SILAC-based relative quantitation. These populations were sorted into the four subpopulations, P1 – P4, as described above. A third population was cultured using

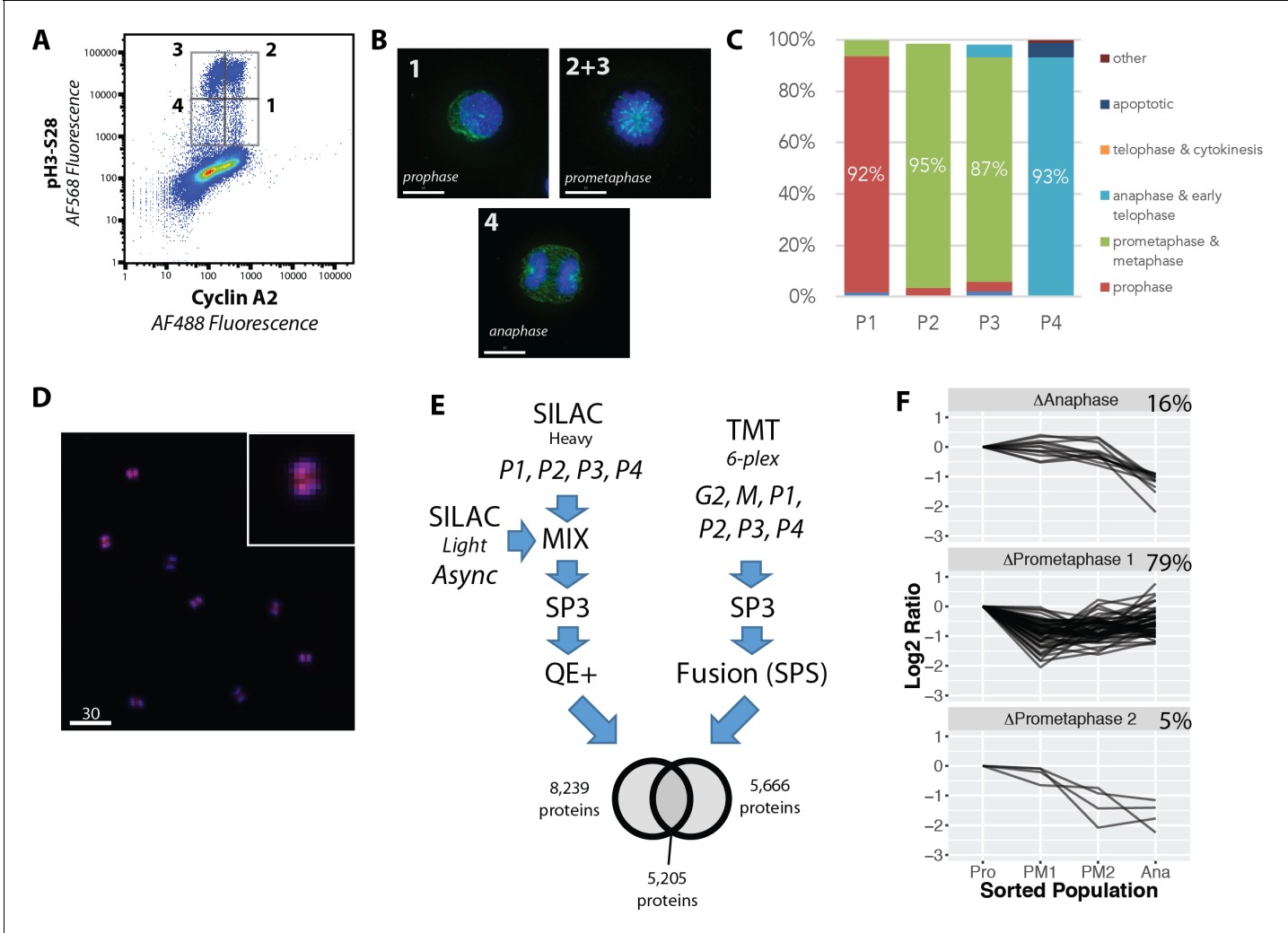

**Figure 8.** Proteome-wide analysis of protein abundance changes between mitotic subphases. (**A**) Flow cytometry analysis of NB4 cells immunostained for H3S28ph and CycA. Gates show populations collected by FACS. (A, right) (**B**) Representative light microscopy images of cell fractions. Scale bars = 10 micron. (**C**) The frequency of each intra-mitotic stage was counted and quantified with 100 cells or more. (**D**) Wide field of view of population 4, the anaphase-enriched population. (**E**) Workflow for MS-based proteomic analysis involving SILAC and TMT based labelling and three biological replicates, resulting in 8,700 proteins identified in total. (**F**) K-means clustering of profiles were qualitatively agglomerated into three groups based on subpopulation where 'trough' in abundance occurs.
DOI: https://doi.org/10.7554/eLife.27574.015
The following figure supplement is available for figure 8:

**Figure supplement 1.** Sorting strategy.
DOI: https://doi.org/10.7554/eLife.27574.016

regular culture media containing non-dialysed bovine serum. For this third experiment, two other subpopulations were collected, that is, G2 cells (i.e. 4N DNA content and pH3-negative cells) and M-phase cells (4N DNA content and pH3-positive cells), in addition to the P1 – P4 populations. Relative quantitation was performed using 6-plex peptide TMT labelling (*Thompson et al., 2003*). To maximise peptide recovery for MS analysis, the SP3 paramagnetic bead cleanup method was used (*Hughes et al., 2014*). SILAC samples were analysed using a Q-Exactive Plus, as previously described (*Endo et al., 2017*). TMT samples were analysed using a Fusion Tribrid instrument, using synchronous precursor selection (SPS) (*McAlister et al., 2014*) to minimise ratio compression. Thus, two distinct relative quantitation strategies were employed to reduce the likelihood that any potential artefacts from a single quantitation strategy would systematically bias the dataset.

Quantitative data for >8,700 proteins were obtained in the combined dataset across three biological replicates. 5,205 proteins (60%) were detected in both the TMT and SILAC datasets. To identify proteins whose abundance changed significantly, protein profiles were calculated based on the mean of the three biological replicates and ratios calculated, relative to prophase levels. Proteins showing missing values in any of the four mitotic subpopulations were discarded, leaving 5,340 proteins. A maximum fold change was calculated and cutoffs were established using Z-scores (~1.8 fold at 95% confidence), which identified 235 proteins meeting this cutoff. Positive correlation between any two biological replicates was used as a second criterion for significance, with 136 proteins meeting these criteria (*Figure 8E*). Several proteins were excluded from the analysis due to missing values, which is a known technical issue with data-dependent acquisition. However, an alternative, non-mutually exclusive explanation, is the lack of detection reflects that the levels of these proteins fall below the detection limit due to physiological down-regulation. We reasoned that the latter explanation is more likely when proteins show missing values reproducibly in a single fraction. Two proteins met this criterion and were added to the candidate list (*Supplementary file 4*), which now totals 138 proteins. However, these two proteins were excluded in further clustering analysis due to missing values being incompatible with the standard k-means clustering algorithm. Most proteins measured (n = 5,202, ~98.5%) showed no significant change (*Figure 8E*).

The mean profiles for the 136 proteins showing the most significantly changing abundance levels were clustered using k-means, with the number of initial clusters (n = 12) determined using a within-cluster sum-of-squares analysis. Because we were interested in identifying candidates for targeted protein degradation during mitosis, we focused on clusters where a decrease in protein abundance was evident. These were manually agglomerated into three clusters, based on the phase in which the decrease is observed (*Figure 8F*). Most proteins (79%), show a decrease in abundance coincident with the CycA +prometaphase subpopulation (PM1), with fewer proteins (16%), decreasing in anaphase and fewer still (5%) decreasing in the CycA- prometaphase subpopulation (PM2).

We next examined the mean protein abundance profiles for cyclins A and B (*Figure 9A,s*.d. shown as a gray ribbon). As expected from the flow cytometry analysis and the sorting strategy employed, cyclin A2 shows high levels in Pro and PM1 and a marked decrease in PM2 and Ana. Two isoforms of cyclin B are detected (B1 and B2). The abundances of both isoforms remain relatively constant between Pro, PM1 and PM2 and decrease in Ana to 25–35% of prophase levels. These data are consistent with the targeting of cyclin B for degradation by the APC/C at the metaphase to anaphase transition. In contrast, the abundance of GAPDH, a protein that is not expected to be targeted for degradation during mitosis, is unchanged between these subpopulations.

## RRM2 is degraded during prometaphase via a MLN-4924-sensitive proteasomal pathway

We were interested in examining other proteins that co-clustered with the cyclin proteins. In the literature, there are few examples of substrates targeted for degradation during prometaphase, as compared with anaphase. We are aware of only two proteins, cyclin A2 and Nek2, for which there is significant evidence in the literature for targeted degradation during prometaphase (*van Zon and Wolthuis, 2010*). Our data show three additional proteins clustering together with cyclin A2, that is, ATAD2, GMNN and RRM2. Inspection of the protein abundance profiles show that while GMNN levels decrease during prometaphase to ~60% of prophase levels, a second major decrease in GMNN abundance occurs during anaphase, where its levels drop to ~20%. ATAD2, a protein involved in transcriptional co-activation of cell cycle genes, such as c-myc, cyclin D1 and E2F1, shows ~50% reduction during prometaphase. ATAD2 contains a conserved, canonical D-box motif situated in a

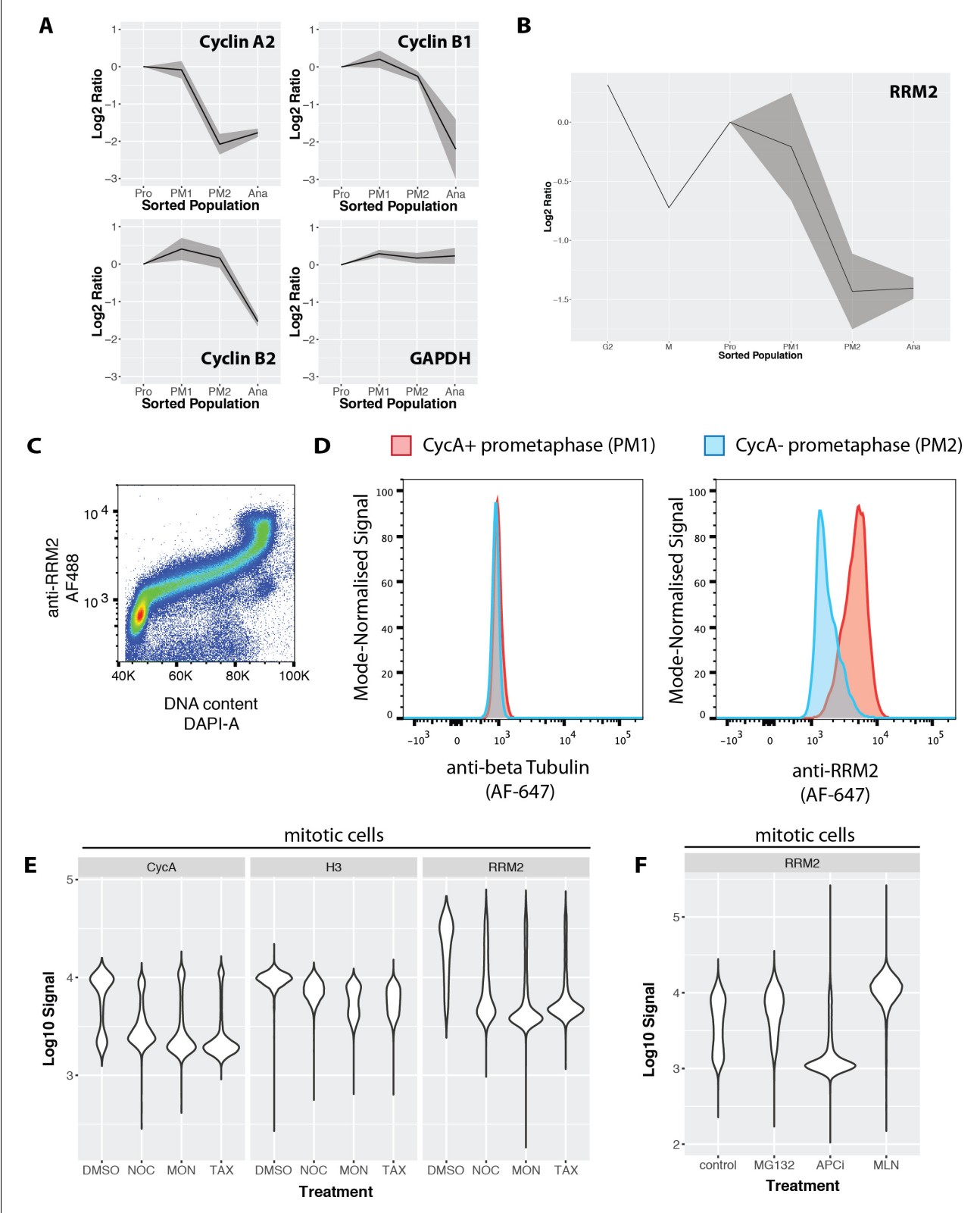

**Figure 9.** Ribonucleotide reductase M2 (RRM2) is degraded during prometaphase in a proteasome-dependent, MLN-4924-sensitive manner. (**A**) Line graphs showing mean abundance profiles for Cyclin A2 (CycA), Cyclin B1, Cyclin B2, and GAPDH. Grey ribbons indicate 1 standard deviation from the mean. (**B**) Mean abundance profile for RRM2. (**C**) Flow cytometry analysis RRM2 levels vs. DNA content. (**D**) Flow cytometry-based comparison of beta-tubulin (negative control, left) and RRM2 (right) levels in CycA+ (red) vs. CycA- (blue) prometaphase cells. (**E**) Violin plots showing CycA, histone H3, and

*Figure 9 continued on next page*

Figure 9 continued

RRM2 levels in cells treated with either DMSO or microtubule drugs that activate the spindle assembly checkpoint (nocodazole, monastrol, taxol). (F) Violin plots showing levels of RRM2 in cells treated with either vehicle control (DMSO), MG132, apcin + proTAME, or MLN4924.

DOI: https://doi.org/10.7554/eLife.27574.017

The following figure supplements are available for figure 9:

**Figure supplement 1.** Validation of the anti-RRM2 antibody used in this study.

DOI: https://doi.org/10.7554/eLife.27574.018

**Figure supplement 2.** RRM2 is cell cycle regulated in U2OS cells, showing peak abundance in G2 phase.

DOI: https://doi.org/10.7554/eLife.27574.019

**Figure supplement 3.** RRM2 abundance decreases during early mitosis in U2OS cells.

DOI: https://doi.org/10.7554/eLife.27574.020

disordered linker region between a predicted DNA-binding bromo- domain and a second globular domain. ATAD2 was also recently shown to be cell cycle regulated via co-immunostaining of FUCCI-expressing cells with a rabbit polyclonal antibody recognising ATAD2 (Human Protein Atlas, www.proteinatlas.org).

We investigated further the observed down-regulation of RRM2. RRM2 is an essential (*Wang et al., 2015*), cell cycle regulated subunit of ribonucleotide reductase that regulates the cellular deoxyribonucleotide pool (*Nordlund and Reichard, 2006*). Properly timed degradation of RRM2 has been suggested to be important, because disrupting the normal degradation timing for RRM2 leads to genomic instability (*D'Angiolella et al., 2012*). RRM2 has been shown to be targeted for degradation by the APC/C-Cdh1 (*Chabes et al., 2003*) and Cyclin F/SCF (*D'Angiolella et al., 2012*) complexes during anaphase/G1 and G2, respectively. Therefore, we were surprised to observe significant decreases in RRM2 abundance in prometaphase fractions, coincident with degradation of cyclin A2 (*Figure 9B*).

A rabbit polyclonal antibody recognising human RRM2, generated by the Human Protein Atlas, was evaluated for specificity by either siRNA depletion of RRM2, or overexpression of a RRM2-GFP construct. (*Figure 9—figure supplement 1*). As shown by both flow cytometry and western blot analysis, depletion of RRM2 significantly reduces or abolishes the antibody signal, and GFP and anti-RRM2 staining patterns show high correlation by immunofluorescence. Combined immunohistochemistry and RNA-Seq analysis across multiple human tissues show that antibody staining is correlated with the cognate RRM2 mRNA (Human Protein Atlas, http://www.proteinatlas.org/ENSG00000171848-RRM2/tissue). This validated RRM2 antibody was used to measure the levels of RRM2 across the cell cycle in both NB4 and U2OS cells. Consistent with its degradation by APC/C-Cdh1, flow cytometry analysis of RRM2 levels and DNA content in NB4 cells show that RRM2 levels are low during G1 phase and increase significantly during S-phase, reaching a peak in G2 and M (*Figure 9C*). Consistent with the data in NB4 cells, immunostaining in FUCCI U2OS also shows that RRM2 abundance peaks in G2 phase (*Figure 9—figure supplement 2*). Interestingly, in both these analyses, RRM2 levels do not show a significant decrease in G2 (*D'Angiolella et al., 2012*). Co-immunostaining cells to detect both cyclin A and H3S28ph enables a flow cytometry-based comparison of RRM2 levels, using the same gating strategy as the MS-based analysis used to define subpopulations. A separate control sample, using an antibody recognising alpha tubulin, was included as a negative control. *Figure 9D* shows the fluorescence signal distributions for two subpopulations marked by H3S28ph and CycA staining: that is, CycA+ prometaphase (PM1) versus CycA- prometaphase (PM2) cells. Comparing protein levels in the respective PM1 vs PM2 subpopulations, alpha tubulin shows no change, as expected (*Figure 9D*, left). In contrast, the level of RRM2 in PM2 is markedly decreased, as compared with PM1, consistent with the MS-based quantitation (*Figure 9D*, right). RRM2 levels during mitosis were independently measured in U2OS cells by co-immunostaining with Cyclin B1 and RRM2 antibodies (*Figure 9—figure supplement 3*). A decrease in RRM2 abundance is observed upon mitotic entry, as measured by nuclear translocation of Cyclin B1. From these data, we conclude that RRM2 levels decrease during early mitosis in both NB4 and U2OS cells.

We next explored potential mechanisms leading to targeted degradation of RRM2. Like CycA, the prometaphase decrease in RRM2 levels occurs in the presence of an active spindle assembly checkpoint (SAC) (*Figure 9E*). Mitotic degradation of RRM2 is sensitive to 4 hr of MG-132 treatment,

suggesting that the degradation occurs via the proteasome pathway. Inhibition of APC/C activity, using combined treatment with apcin and proTAME (*Sackton et al., 2014*), was sufficient to block CycB degradation, but insufficient to stop CycA and RRM2 degradation in NB4 cells. In contrast, treatment with MLN-4924 (*Soucy et al., 2009*), an inhibitor of NEDDylation and cullin ring ligases (CRLs), completely blocks the decrease of RRM2 levels in prometaphase and leads to levels either similar, or even slightly higher, than observed in control cells (*Figure 9F*). As expected, the decrease in CycA in prometaphase is largely unaffected in MLN-4924 treated cells (data not shown).

We conclude that RRM2 is targeted for proteasomal degradation during early mitosis via a MLN-4924-sensitive pathway, probably involving an SCF complex. As discussed further below, a likely candidate is the cyclin F/SCF complex, as RRM2 has been described previously as a substrate of this E3 ligase (*D'Angiolella et al., 2012*).

To maximise community access to the entire dataset described in this study, including the measurements of protein accumulation across the cell cycle, the proteomics data are freely available via several outlets, including the Encyclopedia of Proteome Dynamics (http://www.peptracker.com/epd/) (*Brenes et al., 2017*) (*Figure 10*). The EPD provides a searchable, open access database containing also proteome measurements from multiple large-scale experiments on human cells and model organisms (*Larance et al., 2013*). The data are also available at various stages of analysis, including raw MS files and MaxQuant-generated output (ProteomeXchange) and analysed data (Supplementary files).

## Discussion

A major challenge with the biochemical analysis of mitotic cells is that each sub-phase of mitosis is exceedingly short, lasting only minutes (*Sullivan and Morgan, 2007*). Additionally, there is significant cellular heterogeneity in phase dwell times. Timing heterogeneity, which can be accounted for in timelapse imaging studies (*Akopyan et al., 2014*), present major technical challenges to synchronisation-based strategies for biochemical analysis. In this study, we characterise the PRIMMUS method, which provides a flexible approach that facilitates quantitative proteomic studies on specific cell subsets isolated by FACS based on staining for *intracellular* antigens. For separation of interphase cells (G1, S, G2), centrifugal elutriation is an alternative to FACS, but provides lower resolution

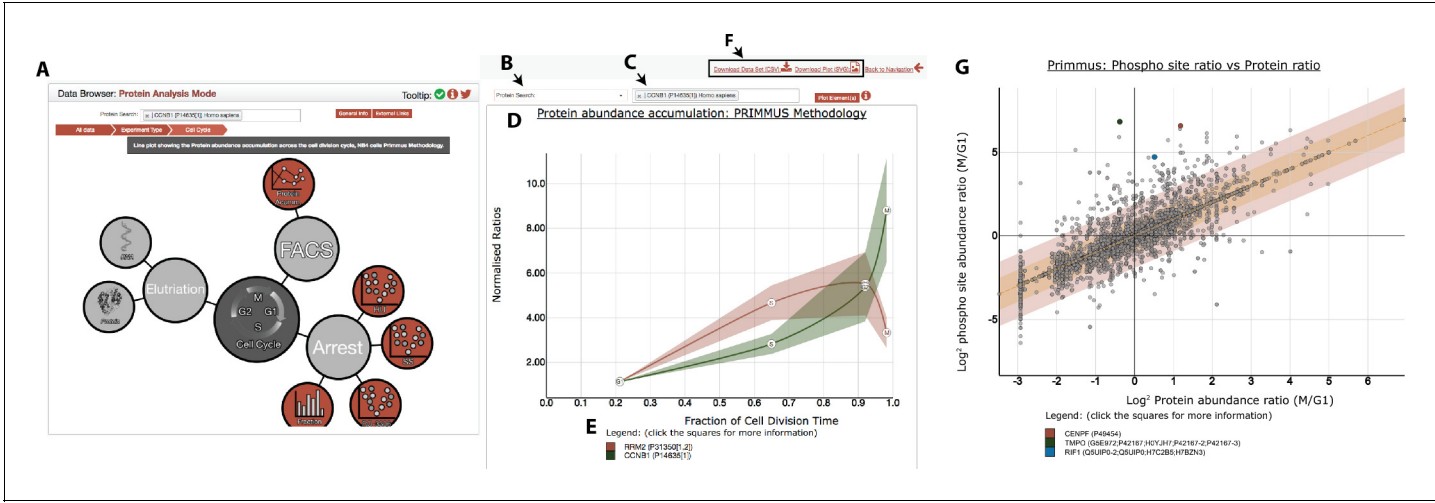

**Figure 10.** The PRIMMUS cell cycle data is accessible through the Encyclopaedia of Proteome Dynamics (EPD). A) Data from proteomic datasets from the Lamond laboratory can be easily visualised for the same proteins using the navigation bubble map. User interface features include: B) specifying type of search, including search for individual proteins, GO term, and CORUM complex membership, (C) an input box for protein and other identifiers (e.g. GO term), (D) a ribbon graph showing plots for input protein identifiers (here for illustration are shown RRM2 and CCNB1) with lines indicating mean profile and ribbons indicating s.e.m., (E) an interactive legend to show more information on individual proteins, and F) options to output visualisation as an SVG file or underlying data in CSV format. (G) An example plot correlating protein abundance and phosphorylation changes. Several proteins containing 'early rising' phosphorylation sites are highlighted: CENPF, TMPO, and RIF1.
DOI: https://doi.org/10.7554/eLife.27574.021

separation, is not applicable to all cell types and does not efficiently separate G2 and M phase cells. We therefore used FACS to produce highly enriched populations of cells at specific cell cycle stages. Cells growing in asynchronous cultures were FACS separated by either, a) DNA content and phosphorylation of histone H3, obtaining high purity populations of G1, S, G2 and M phase cells, or by b) DNA content, phosphorylation of histone H3 and the degradation of CycA, obtaining high purity populations of prophase, prometaphase and anaphase intra-mitotic cells. Using these isolated cell populations, we provide the first specific MS-based proteomic analysis of intra-mitotic phase cells isolated from asynchronously growing cultures.

We validated the PRIMMUS method by demonstrating that global MS-based protein identification and quantitation is compatible with the analysis of populations of fixed cells that have been permeabilised, stained to detect *intracellular* antigens and isolated by FACS. While FACS has been used previously in conjunction with RNA-seq to compare mRNA abundances of cell subsets (*Hrvatin et al., 2014*), this study provides the first example we know of where permeabilised, fixed and intracellular immunostained cells have been FACS sorted and used for quantitative, MS-based proteome analysis. In principle, the PRIMMUS approach can be used to characterise any distinct type of cell subpopulation that can be defined using one or more diagnostic antigens, an abundance differential for a specific epitope, or combination of epitopes, including intracellular and intranuclear antigens. We also show that PRIMMUS enhances the sensitivity of quantitative proteomics technology to detect either changes in abundance, and/or changes in other protein properties, such as post-translational modifications, because it facilitates the analysis of the specific subsets of cells in which the change occurs, without diluting this signal by analysing mixed populations, including non-responding cells. This is illustrated here by our demonstration of up to a five-fold sensitivity gain in detecting cell cycle regulated protein abundance changes, as judged by comparing data obtained using PRIMMUS, with data from cells isolated using centrifugal elutriation.

We have recently shown by proteomic analysis of NB4 cells arrested at specific cell cycle stages by drug treatments that the stress resulting from drug arrest causes major changes in the proteome distinct from the physiological regulation of protein levels during unperturbed cell cycle progression (*Ly et al., 2015*). In the PRIMMUS method, fixation by FA captures cells 'frozen in motion' and thereby prevents significant perturbation of cellular physiology due to extended sample handling. Consistent with this idea, we previously showed that that the abundance of the apoptosis-associated protein BID is upregulated by the CDK1 inhibitor RO-3306, which arrests cells at the G2 and M border (*Ly et al., 2015*). In this study, using PRIMMUS, we observe BID levels stay relatively constant across the unperturbed cell cycle, consistent with our results comparing minimally perturbed cells separated by centrifugal elutriation (*Ly et al., 2014*). While FACS can also be used to sort live cells, thereby avoiding procedures involving fixation and permeabilisation, the process of FACS-based separation of live cells can induce cellular stress and this in turn can rapidly alter the proteome. The proteomes of live cells immediately after FACS may thus be significantly remodeled by activation of cellular stress responses. Furthermore, most live cell strategies require the construction of cell lines expressing one or more fluorescent-tagged fusion proteins. In contrast, the PRIMMUS approach described here avoids any requirement for the use of exogenous, tagged proteins and provides a general strategy that can be applied to both cultured cell lines and primary cells isolated ex vivo.

We used PRIMMUS, combined with high accuracy cell mixing and quantitative metabolic labelling, to measure and compare the abundances of thousands of individual proteins across an unperturbed cell division cycle, including different substages of mitosis. Our data show that while exponential accumulation is observed for bulk protein, the rate of accumulation of individual proteins can deviate significantly from an exponential growth fit. For example, histones, which constitute ~3–5% of the total protein abundance in NB4 cells (*Ly et al., 2014*), deviate from bulk protein accumulation by remaining flat between G2 and M, consistent with the synchronisation of DNA replication and histone synthesis (*Nelson et al., 2002*), (*Robbins and Borun, 1967*). Intriguingly, for p53 and some other proteins associated with stress responses and/or apoptosis, protein abundance remains relatively flat across the cell division cycle. Due to cell cycle-dependent increases in cell size and bulk protein abundance, the effective net intracellular concentrations of these proteins must thus decrease as the cell cycle progresses. Any functional consequence of a decrease in their abundance may however be offset if mechanisms exist to create local concentration hotspots at specific subcellular locations.

Detailed proteomic analysis of protein abundances across prophase, prometaphase and anaphase cells showed that only ~1.5% of the proteins quantitated significantly decrease in abundance. These proteomic data showed RRM2 levels decreasing during prometaphase. Consistent with this, subsequent flow cytometry analysis showed that RRM2 and CycA levels are correlated during mitosis. In U2OS cells, RRM2 levels degrade in early mitosis, either coincident with, or sometime shortly after the appearance of cyclin B1 in the nucleus.

The decrease in RRM2 levels was prevented by treatment of cells with either MLN-4924, or MG-132, consistent with the mechanism causing a decrease in RRM2 levels involving a cullin E3 ubiquitin ligase and proteasomal degradation. It has been shown that RRM2 is targeted for degradation by Cyclin F/SCF during G2 (*D'Angiolella et al., 2012*) and several studies have shown co-immunoprecipitation of RRM2 and Cyclin F in asynchronous cell extracts (*D'Angiolella et al., 2012*; *Huttlin et al., 2015*; *Hein et al., 2015*). It has also been reported that disruption of the Cy motif in RRM2, which is important for recognition by Cyclin F/SCF, stabilises RRM2 levels and increases DNA pools and genome mutation frequencies (*D'Angiolella et al., 2012*). It will be interesting in the future to investigate further the timing of RRM2 degradation and its relation to maintaining genome stability.

We show that PRIMMUS is compatible also with analysis of cell cycle regulated protein phosphorylation by comparing phosphorylation in G1-, S-, G2- and M-phase enriched fractions purified by FACS. While essentially all significantly changing phosphorylation sites peak during mitosis, we identify a subset of phosphorylation sites that show increased phosphorylation in the G2-enriched fraction, which we call 'early risers'. These early risers share functional similarities, including enrichment in the 'classic' CDK substrate phosphorylation motif and localisation to chromatin and the nuclear envelope. We suggest that these early rising sites may be the downstream effect of mitotic entry kinase activity, which includes contributions from Plk1 (*Smith et al., 2011*), Cdk1(*Gavet and Pines, 2010*), and Cyclin A-CDK2 (*Gong et al., 2007*). Interestingly, the sites that change by the highest amount in G2, besides factors strongly associated with DNA replication, are localised to the nuclear pore and nuclear envelope. Additionally, several sites, such as on lamina-associated polypeptide 2 (LAP2) and NUP153, contain both Plk1 and Cdk substrate sequence motifs (*Supplementary file 2*). Proteins at the nuclear envelope, including lamin B and NUP153, have been shown to be likely candidate substrates for Cyclin A/CDK2 (*Chi et al., 2008*). Evidence from immunoprecipitation experiments suggest that these proteins may not be exclusively Cyclin A/CDK2 targets, as there is significant overlap in proteins immunoprecipitated with Cyclin A vs. Cyclin B (*Pagliuca et al., 2011*). That 'early rising' phosphorylations show significant increase in the mitotic fraction is consistent with increasing CDK activity for the same substrates in the transition from from cyclin A/CDK2 to cyclin B/CDK1 (*Uhlmann et al., 2011*) (*Stern and Nurse, 1996*). An expanded analysis of phosphorylation during S- and G2-phases will be an important goal of future studies and will help to define the targets of the mitotic entry network in detail. This may provide clues to the sequence of signaling events that precede nuclear envelope breakdown. Future studies will also explore phosphorylation changes between mitotic subphases, which will help elucidate the role of mitotic kinases and phosphatases during progression through mitosis.

When the phosphoproteomic dataset from this study was compared with a previously reported atlas of mitotic phosphorylation in human cells derived using nocodazole arrested HeLa cells (*Olsen et al., 2010*), some phosphorylation sites were identified that changed in one dataset, but not the other. The sites that differed between the two datasets were preferentially on proteins with specific functional GO annotations, including 'cell cycle' and 'mitosis' for phosphorylation sites that are changing exclusively in the mitotic fraction in this PRIMMUS dataset, but instead proteins with phosphorylation sites that are changing exclusively in the mitotic fraction in the previously reported HeLa dataset are enriched for the annotation 'splicing' and not 'cell cycle' or 'mitosis'. Several differences between these studies could explain study-specific phosphorylation changes. For example, cell-type specificity in cell cycle regulated protein phosphorylation, for example NB4 cells (this study), as compared with HeLa cells used in the previous study. An alternative, non-mutually exclusive, explanation could be the difference in the methods used for obtaining mitotic cells in the respective studies, that is FACS (this study) vs nocodazole-arrest (HeLa dataset [*Olsen et al., 2010*]). Previous work has shown that cell cycle arrests can induce proteomic changes that are not observed in a minimally perturbed cell cycle (*Ly et al., 2015*). Further analysis will be required to investigate the basis for the observed study-specific phosphorylation changes.

We show that for the TPX2 protein, the phosphorylation increase observed during G2 and mitosis has functional consequences on spindle formation. Specifically, mutation of the early rising phosphorylation site, TPX2 S738, to either a non-phosphorylatable (alanine) or a phosphomimetic (aspartic acid) residue, elicits defects in bipolar spindle assembly. The 738 residue lies within a critical C-terminal domain of TPX2 that has been shown to be important both in recruiting the mitotic kinesin Eg5 to spindle microtubules in cells and in modulating Eg5-dependent microtubule gliding activity in vitro (*Ma et al., 2011*; *Eckerdt et al., 2008*). The phenotypes of the phosphomimetic and non-phosphorylatable mutations at S738 in TPX2 are not as severe as loss of the entire C-terminal domain (*Ma et al., 2011*) suggesting that phosphorylation at this site modulates, but does not abolish, the interaction with Eg5. A variety of defects were observed for each mutant, including monopolar and multipolar spindles, suggesting defects in motor-dependent force generation during spindle formation.

Prior work has shown that Eg5 is transported poleward in a dynein-dependent manner during early mitosis and reverses direction in anaphase, moving away from the spindle poles (*Gable et al., 2012*; *Uteng et al., 2008*). However, in cells depleted of TPX2, Eg5 moves away from the poles, towards microtubule plus-ends in early mitosis, showing that TPX2 is needed to couple Eg5 to dynein. Thus, the dynamics of Eg5 change in a TPX2 and cell cycle regulated manner, consistent with a regulated interaction between the two proteins.

Because the TPX2-738 phosphorylation site was classified as an early riser, an early mitotic phenotype might be expected. TPX2 contains a functional NLS and is nuclear throughout interphase; a small fraction of the protein has been detected at the centrosome, in prophase, in mammalian cells (*Ma et al., 2011*). Evidence in plant cells suggest a role during early prophase (*Vos et al., 2008*). In these cells, TPX2 is detected on microtubules outside of the prophase nucleus and microinjection of anti-TPX2 antibodies during prophase inhibits nuclear envelope breakdown. In mammalian cells, the weak signal from cytoplasmic TPX2-GFP at prophase and the small number of prophase cells precluded detecting any possible phenotype in the present experiments. It should be noted that other work has identified nuclear roles for TPX2 in the DNA damage response (*Neumayer et al., 2014*) so it remains possible that phosphorylation of TPX2 regulates multiple events throughout the cell cycle.

In addition to PTM analyses, the PRIMMUS workflow can be extended in future by combining it with other complementary approaches to extend the depth of proteomic analysis of the selected cell subpopulations beyond only abundance measurements, for example, using MS-based techniques for identifying protein interaction partners and protein complexes (*Kirkwood et al., 2013*; *Kristensen et al., 2012*). PRIMMUS is highly complementary to recent studies dissecting, in high resolution, the timing of mitotic kinase activities and quiescence control by automated fluorescence imaging of fixed, immunostained cells (*Akopyan et al., 2014*; *Spencer et al., 2013*). As FA fixed cells are compatible with downstream RNA-seq analysis and ChIP studies, combining proteomics on sorted cells also with these high-throughput approaches would expand the in-depth analysis of gene expression (transcriptome and proteome) of rare populations and/or biochemically well-defined subpopulations, such as specific hematopoeitic cell populations that cannot be separated on the basis of cell surface markers alone.

# Materials and methods

## Key resources table

| Reagent type (species) or resource | Designation | Source or reference | Identifiers | Additional information |
|---|---|---|---|---|
| Human Cell Line | NB4 | Ron Hay lab (Dundee), *Lanotte et al., 1991* | RRID:CVCL_0005 | Tested negative for mycoplasma (Lonza MycoAlert) |
| Human Cell Line | HeLa | ATCC | RRID:CVCL_0030 | Tested negative for mycoplasma |
| Human Cell Line | U2OS | ATCC | RRID:CVCL_0042 | Tested negative for mycoplasma |
| Porcine Cell Line | LLC-Pk1 | ATCC | RRID:CVCL_0391 | Tested negative for mycoplasma |
| Antibody | rat anti-tubulin YL1/2 | Accurate Scientific | RRID:AB_2687885 | |
| Antibody | rabbit anti-RRM2 | Human Protein Atlas | RRID:AB_2683304 | |

*Continued on next page*

*Continued*

| Reagent type (species) or resource | Designation | Source or reference | Identifiers | Additional information |
|---|---|---|---|---|
| Antibody | mouse anti-H3S10ph | Cell Signaling Technology | RRID:AB_331748 | |
| Antibody | rat anti-H3S28ph HTA28 | Abcam | RRID:AB_2295065 | |
| Antibody | mouse anti-tubulin (DM1a) | Sigma | RRID:AB_477593 | |
| DNA, Expression construct | RRM2-GFP | Origene | RG205718 | |

## Cell culture

The NB4 cell line (RRID: CVCL_0005) was originally established from acute myeloid leukemia blast cells grown on bone-marrow stromal fibroblasts (*Lanotte et al., 1991*). NB4 cells were obtained from the Hay laboratory (University of Dundee). Cells were cultured at 37°C in the presence of 5% $CO_2$ as a suspension in RPMI-1640 (Life Technologies, UK) supplemented with 2 mM L-glutamine, 10% v/v foetal bovine serum (FBS, Life Technologies), 100 units/ml penicillin and 100 μg/ml streptomycin (100X stock, Life Technologies). Cell cultures were maintained at densities between $1 \times 10^5$ to $1 \times 10^6$ cells/ml. For SILAC labelling, culture media without arginine or lysine (Dundee Cell Products, UK) was supplemented with either 'light' (Arg0, Lys0, Cambridge Isotope Labs, Tewksbury, MA, USA) or 'heavy' (Arg10, Lys8, Cambridge Isotope Labs) isotopomers of arginine and lysine. SILAC labelling media was additionally supplemented with dialysed serum (10% final), 1X insulin-transferrin-selenium (100X stock from Life Technologies), 1x MEM vitamins (100X stock from Life Technologies), and 90 mg/l proline. The U2OS cell line (U2OS; RRID: CVCL_0042) was cultivated at 37°C in a 5% $CO_2$ humidified environmental condition using McCoy's 5A modified medium supplemented with 10% foetal bovine serum (FBS, Sigma, St Louis, MO, USA) and 1% L-glutamine. Cells were harvested at around 60% confluency using trypsin (Trypsin-EDTA solution from Sigma-Aldrich). HeLa cells (ATCC, RRID: CVCL_0030) were cultured in DMEM media supplemented with 10% FBS (Gibco, South American origin) and penicillin/streptomycin. For transfection experiments, lipid complexes were prepared in serum-free media (Opti-MEM, Gibco)

## TPX2 depletion and rescue experiments

LLC-Pk1 cells (CVCL_0391) were grown in Hams F-10 mixed one to one with Opti-MEM, and containing 7.5% foetal calf serum and antibiotic/antimycotic at 37°C and 5% $CO_2$. Prior to use in experiments cells were plated on $22 \times 22$ mm #1.5 coverslips in 35 mm dishes or on to the surface of glass bottom petri dishes (Mattek Corp).

To generate mutants at residue S738, a mouse bacterial artificial chromosome (BAC) expressing a localisation and affinity purification tagged TPX2 was used (referred to hereafter as GFP tagged). Mutations were generated using site directed mutagenesis in bacteria and confirmed by sequencing (*Poser et al., 2008*). Purified BAC DNA was nucleofected into LLC-Pk1 cells using an Amaxa nucleofector and Mirus nucleofection reagent and antibiotic selection was performed as previously described (*Ma et al., 2011*). Prior to use in experiments, the cells were nucleofected with siRNA targeting endogenous pig TPX2 (5' GAAUGGUACAGGAGGGCUU 3'). As a control for nucleofection, cells were nucleofected with siGLO (Dharmacon, Lafayette, CO, USA) according to the manufacturers recommendation. To score mitotic morphology, live cells were imaged on a confocal microscope in the GFP channel or following fixation (3.7% paraformaldehyde, 0.1% glutaraldehyde in PBS containing 0.5% Triton X-100) and staining for microtubules using a mouse anti-tubulin antibody (DM1a, RRID: AB_477593) or rat anti-tubulin (YL1/2, RRID: AB_2687885) and appropriate secondary antibodies. Incubations with primary antibodies were performed for either 1 hr at 37°C or overnight at room temperature; secondary antibodies were used at room temperature for 45 min. Antibodies were mixed at the appropriate final dilution with PBS containing 2% BSA, 0.1% Tween and 0.02% sodium azide. Cells were imaged using a Nikon Eclipse TE300 or TiE equipped with a Yokagawa spinning disc scan head and 100X NA 1.4 objective lens as previously described (*Ma et al., 2011*). Image analysis was performed in ImageJ or FIJI. For long-term imaging, cells were imaged on a Nikon TiE with a spinning disc confocal system (Yokogawa) at 40X. Cells were maintained at 37°C

and 5% $CO_2$ using an Oko Lab environmental stage insert. Images were collected from multiple points at 3 min intervals; images were collected at 488 nm or both the 488 and 561 nm for 3–5 hr. To measure mitotic duration, the time between nuclear envelope breakdown and anaphase was measured from the 488 channel.

## Immunofluorescence staining for FACS

NB4 cells (~0.5 $\times$ $10^8$ cells) were washed once with phosphate-buffered saline (PBS) and resuspended in freshly prepared 50 ml 0.5% formaldehyde in PBS. Cells were fixed with formaldehyde for 30 min at room temperature with shaking. Cells were pelleted, and permeabilised with 50 ml cold 90% methanol. Cells were then stored at −20°C prior to staining.

Fixed, permeabalised cells were washed once with PBS and resuspended in blocking buffer, 5% bovine serum albumin (BSA) in Tris-buffered saline (TBS) + 0.05% sodium azide. Cells were blocked for 10 min at room temperature, pelleted, and resuspended in primary antibody solution (1:200 in blocking buffer). Cells were stained with primary antibody overnight at 4°C. The primary antibodies used for immunostaining are rabbit anti-RRM2 (HPA056994, RRID: AB_2683304), mouse anti-H3S10ph (Cell Signaling Technology 9706, RRID: AB_331748), rat anti-H3S28ph HTA28 (Abcam ab10543, RRID: AB_2295065), and mouse anti-alpha tubulin (Sigma DM1a, RRID: AB_477593). Stained cells were then washed twice with PBS, and stained with dye-conjugated secondary antibodies (1:200 in blocking buffer) for 1 hr at room temperature. Stained cells were washed twice with PBS, pelleted, and resuspended in DAPI solution (5 µg/ml in PBS).

## RRM2 depletion and overexpression

HeLa cells were transfected with either siJumble (*Hutten et al., 2014*), or a pool of 4 siRNA targeting RRM2 (ON-TARGETplus siRNA, GE Healthcare L-010379–00) using Lipofectamine RNAiMAX (Life Technologies) using manufacturer's instructions. Cells were incubated 24 hr before harvest for immunoblot and flow cytometry assays. For overexpression experiments, HeLa cells were transfected with either GFP-fibrillarin, or RRM2-GFP (Origene RG205718) using Lipofectamine 3000 (Life Technologies). Transfected cells were incubated for 24 hr before harvesting for immunostaining using mouse anti-GFP (Roche, RRID: AB_390913) and rabbit anti-RRM2 primary antibodies (HPA056994, RRID: AB_2683304).

## Flow cytometric analysis of cell cycle distribution

Cells stained with either PI or DAPI were analysed on an LSR Fortessa flow cytometer and data acquired using DIVA software (Becton Dickinson). DNA content was evaluated based on DAPI fluorescence (measured using 355 nm excitation and emission at 450 ± 50 nm) or PI fluorescence (measured using 488 nm excitation and emission at 585 ± 42 nm). Doublet discrimination was used to remove cell doublets and clumps using DAPI/PI-A and DAPI/PI-W measurements. The cell cycle distribution of single (gated) cells was plotted as DAPI/PI-A. Data was analysed using Flowjo software (Treestar inc.) and cell cycle distributions determined using the Watson/Pragmatic model.

## Fluorescence-activated cell sorting (FACS)

FACS was performed on an Influx cell sorter (Becton Dickinson) equipped with 488 nm, 405 nm, and 642 nm laser light sources. Forward angle light scatter (FSC) and side angle (90°) light scatter (SSC) were determined by detection of scattered 488 nm light. DAPI fluorescence was measured using 405 nm excitation and emission detected at 460 ± 50 nm, Alexa Fluor 488 fluorescence was measured using 488 nm excitation and emission detected at 530 ± 40 nm, Alexa Fluor 568 fluorescence was measured using 488 nm excitation and emission detected at 610 ± 20 nm, and Alexa Fluor 647 fluorescence was measured using 642 nm excitation and emission detected at 670 ± 30 nm.

Signal processing was performed by a 16-bit analogue to digital converter, providing 65,536 channels. Linear scaling (0 to 65,536 channels) or logarithmic scaling (four log decades of 16,384 channels) of data was performed, depending on parameter detected (see below). All parameters are presented as measurements of pulse height, unless otherwise stated. Compensation was not applied due to the careful selection of spectrally distinct fluorophores, minimising spectral overlap, and gating strategies employed.

For cell sorting phosphate buffered saline pH 7.5 (PBS, Sigma Cat no. P-4417) was used as sheath, and sorting was performed using a 100 µm nozzle and sheath pressure of 20 psi. Typical drop drive frequency used was in the region of 40 kHz, enabling a maximum event rate for sorting of 10,000 events per second (i.e. 1 event per four drops). The sorting mode selected was drop count = 1.0, drop attributes = pure (this ensures exact counts of cells collected and sort purity).

Immediately prior to sorting, cells were passed through a 50 µm filter (Filcons, Becton Dickinson Cat no. 340629) to remove cell clumps. Sorted cells were collected into either 1.5 ml or 2.0 ml Protein LoBind Eppendorf tubes containing ~50 µl of PBS. Where cell numbers allowed, post-sort purity was checked by reanalysis of the collected cells by flow cytometry, and found to be >96% in all cases. Sorting and data analysis during sorting was performed using FACS Sortware (Becton Dickinson) and post-sort data analysis performed using Flowjo.

## FACS population identification and gating strategies

To separate interphase and mitotic cells, cells were distinguished from particulate material based on FSC and SSC (Suppl. *Figure 1A*). DAPI fluorescence was detected and presented on a linear scale, setting the G1 peak at approximately channel 20,000 to allow good resolution of cell cycle, while keeping all data on scale. Integration of the DAPI signal pulses was performed electronically to provide pulse area (A) and pulse width (W), as well as pulse height (H). Cell doublets and aggregates were excluded based on DAPI-W v DAPI-A measurements. G1, S and G2 phases of the cell cycle were identified based on DNA content, determined from histogram plots of DAPI-A. In order to minimise contamination of samples with other phases of the cell cycle, the bottom half of the G1 peak, top half of the G2 peak and middle of the S phase were collected (Suppl. *Figure 1A*). H3S10ph positive cells were identified using either AlexaFluor 488 (H3S10ph)-conjugated antibodies. In both cases, fluorescent parameters were scaled logarithmically and positive gates for phosphohistone H3 determined using a negative (minus antibody) control. Representative plots of the gating strategy used is shown in Suppl. *Figure 1A*.

To separate mitotic subphases, interphase and mitotic cells were first identified based on DNA content determined by DAPI staining as described above. Rat anti-H3S28ph antibody detected by anti-Rat conjugated AlexaFluor 568 and Mouse anti-Cyclin A detected by anti-Mouse conjugated AlexaFluor647 were employed to detect the relevant antigen, and logarithmic scaling used for all two parameters. The full gating strategy is described below, where double carats (>>) indicate the deprecated level in the gating hierarchy. The gating strategy is also shown as flow diagrams in Suppl. *Figure 1B*.

[Parameters used for gating: target subpopulations]

FSC v SSC: Identification of cells and elimination of debris >>DAPI W v DAPI-A: Identification of single 4N DNA content cells and exclusion of doublets and cell aggregates >> AF488 hr (pHH3) v AF568-H (Cyclin A): Sort gates identifying Cyclin A low/H3S28ph mid, Cyclin A low/H3 S28ph high, Cyclin A high/H3 S28ph mid, Cyclin A high/H3 S28ph high

## Fluorescence microscopy

Cells purified by FACS were settled onto poly-lysine-coated coverslips (BD Biosciences) for 30 min at room temperature. The liquid was then carefully aspirated. Cells were fixed with 2% FA in PBS for 10 min at room temperature. Cells were washed twice with PBS and stained with primary antibodies for 1 hr at room temperature. Cells were washed twice with PBS and stained with dye-conjugated secondary antibodies for 30 min at room temperature. Cells were washed twice with PBS and stained with DAPI (5 µg/ml in PBS) for 1 min at room temperature. Cells were washed once with PBS and mounted in Vectashield medium (Vector Laboratories). Cells were visualised using a wide-field fluorescence microscope (Zeiss, Jena, Germany; Axiovert-DeltaVision Image Restoration; Applied Precision, LLC).

LLC-Pk1 cells were cultured in Hams F-10 mixed one to one with Opti-MEM, and containing 7.5% fetal calf serum. Prior to use in experiments cells were plated on 22 × 22 mm #1.5 coverslips in 35 mm dishes or on to the surface of glass bottom petri dishes (Mattek Corp). Cells were imaged using a Nikon A1R + resonant scanning confocal system equipped with a 60X NA 1.4 objective lens. For live cell imaging, cells were imaged using a Nikon Eclipse TE300 or TiE equipped with a Yokagawa

spinning disc scan head and 100X NA 1.4 objective lens as previously described (*Ma et al., 2011*). Image analysis was performed in ImageJ or FIJI.

Immunostaining of the human U2OS cells was performed in a glass bottom plate (Greiner Sensoplate Plus, Cat# 655892, Greiner Bio-One, Germany) coated with 12.5 µg/ml human fibronectin (VWR). Approximately 6000 cells were seeded in each well and incubated at 37°C for 24 hr. After washing with Phosphate buffered saline (PBS), cells were fixed with 40 µl 4% ice cold PFA (Sigma Aldrich) dissolved in growth medium supplemented with 10% serum for 15 min and permeabilised with 40 µl 0.1% Triton x-100 (Sigma Aldrich) in PBS for 3 × 5 min. The primary antibody targeting RRM2 (HPA056994, RRID: AB_2683304), was dissolved to 2 µg/ml in blocking buffer (PBS + 4% FBS) containing 1 µg/ml chicken anti-tubulin (Abcam, ab89984, RRID: AB_10672056) and 2.5 µg /ml mouse anti-CCNB1 (BD Biosciences 610220, RRID: AB_397617). After washing the cells with PBS, diluted primary antibodies were added (40 µl/well) and the plates were incubated at 4°C. After overnight incubation, all wells were washed with PBS for 3 × 10 min. Secondary antibodies, goat anti-rabbit488 (ThermoFisher A11034, RRID: AB_2576217), goat anti-mouse555 (ThermoFisher A21424, RRID: AB_141780) and goat anti-chicken647 (ThermoFisher A21449, RRID:AB_2535866) diluted to 2.5 µg /ml in blocking buffer were added and the plates incubated for 90 min at room temperature. After washing with PBS, all wells were mounted with PBS containing 76.5% glycerol.

U2OS cells were imaged using a Leica SP5 confocal microscope (DM6000CS) equipped with a 63-x/1.4 NA oil immersion objective was used for image acquisition. Images were acquired at room temperature in three sequential steps with the following scanning settings; Pinhole 1 Airy unit, 16-bit acquisition and a pixel size of 80 × 80 nm. The z focus-level was manually adjusted to represent the best visualisation of the target protein. The detector gain was maintained constant across all samples. Mitotic images were selected and the cells were labelled manually.

## Cell lysis, reverse crosslinking, protein precipitation/SP3, Ti:IMAC phosphoenrichment

Cells were resuspended in 4% SDS in PBS, homogenised with a probe sonicator (Branson, 10% power, 20 s, 4°C), and heated to 95°C for 30 min to reverse crosslinks. For MS analysis, proteins were then reduced and alkylated using TCEP (25 mM final concentration, Sigma) and iodoacetamide (55 mM final concentration, Sigma). G1, S, G2, and M phase lysates for the single shot analyses were then chloroform-methanol precipitated.

The interphase fractions for phosphoenrichment and the mitotic substage fractions (G2, M, Pro, PM1, PM2, and Ana) were processed using the SP3 method, as described previously (*Hughes et al., 2014*). The mean cell counts for the phosphoenrichment experiment, which was performed in biological duplicate, was 10, 6, 3.5, and 1.5 million for G1, S, G2, and M phase fractions, respectively. The mean cell counts for mitotic subphases experiment, which was performed in biological triplicate, were 0.14, 0.52, 0.36, and 0.12 million for Pro, PM1, PM2, and Ana fractions, respectively. Lysis buffer volumes were adjusted according to cell count, and identical volumes were used for TMT labelling and mixing. Cell disruption and DNA homogenisation was performed using a Pico Bioruptor (Diagenode). Reverse crosslinking was performed as above. Proteins were recovered from lysates using the SP3 method and digested 'on-bead' with Lys-C followed by Trypsin. Peptides were then recovered by SP3, and TMT labelled according to manufacturer's instructions. For phosphoenrichment, TMT labelled peptides were mixed and subjected to magnetic Ti:IMAC enrichment (Resyn Biosciences) using the manufacturer's protocol.

## Immunoblot analysis

Lysates for SDS-PAGE analysis were prepared in lithium dodecylsulphate sample buffer (Life Technologies) and 25 mM TCEP. Samples were heated to 65°C for 5 min and then loaded onto a NuPage BisTris 4–12% gradient gel (Life Technologies), in either MOPS, or MES buffer. Proteins were electrophoresed and then wet transferred to nitrocellulose membranes at 35 V for 1.5 hr. Membranes were then blocked in 5% BSA in immunoblot wash buffer (TBS +0.1% Tween-20) for 1 hr at room temperature. Membranes were then probed with primary antibody overnight at 4°C, washed and then re-probed with LiCor dye-conjugated secondary antibodies (either IRDye-688 or IRDye-800). Primary antibodies for cell cycle immunoblot analysis were obtained from Cell Signaling Technology (cyclin

B1, cyclin A, cyclin E, CDT1). Bands were visualised using the Odyssey CLx scanner (LiCor Biosciences).

## Offline HPLC and LC-MS/MS analysis

Chloroform-methanol precipitated protein pellets were resuspended in 8 M urea in digest buffer (100 mM Tris pH 8.0, 1 mM CaCl$_2$). The protein solution was then diluted to 4 M urea with digest buffer and then digested with Lys-C, which was added at a 1:50 w/w Lys-C:protein ratio from a 1 mg/ml stock in water (Wako Chemicals) overnight at 37°C. The lysates were then further diluted with digest buffer to 0.8 M urea and digested with trypsin, which was added at a 1:50 w/w trypsin:protein ratio from a 0.2 mg/ml stock in 50 mM acetic acid (Thermo Pierce) for four hours at 37°C. The digests were then desalted using SepPak-C18 SPE cartridges, dried, and resuspended in 5% formic acid. Peptide concentrations were determined using the amine-reactive, fluorogenic CBQCA assay (Life Technologies).

SP3-processed proteins were digested in a similar manner to above. For one biological replicate, the resulting peptides were TMT-labelled according to manufacturer's instructions. Peptides (SILAC or TMT) were then mixed, dried, and resuspended in high pH reverse phase buffer A (10 mM ammonium formate in 2% ACN, pH 9.3). Peptides were then chromatographed on a Dionex Ultimate 3000 off-line HPLC equipped with a XBridge Peptide BEH C18 4.6 mm x 250 mm column packed with 3.5 μm particles containing 130 angstrom pores (Waters) and a standard gradient over 20 min from 25% at 0 min to 60% B at 11 min (10 mM ammonium formate in 80% ACN, pH 9.3) at a flow rate of 1 ml/min. 48 fractions were collected into 24 wells in a concatenated format.

SILAC-labelled peptides were analyzed using a Dionex RSLCnano HPLC-coupled Q-Exactive Orbitrap mass spectrometer (Thermo Fisher Scientific). Peptides were first loaded onto a 2 cm PepMap trap column in 2% acetonitrile +0.1% formic acid. Trapped peptides were then separated on an analytical column (75 μm x 50 cm PepMap-C18 column) using the following mobile phases: 2% acetonitrile +0.1% formic acid (Solvent A) and 80% acetonitrile +0.1% formic acid (Solvent B). The linear gradient began with 5% B to 35% B over 220 min with a constant flow rate of 200 nl/min. The peptide eluent flowed into a nanoelectrospray emitter at the front end of a Q-Exactive Plus (quadrupole Orbitrap) mass spectrometer (Thermo Fisher Scientific). A typical 'Top10' acquisition method was used. Briefly, the primary mass spectrometry scan (MS1) was performed in the Oribtrap at 70,000 resolution. Then, the top 10 most abundant m/z signals were chosen from the primary scan for collision-induced dissociation in the HCD cell and MS2 analysis in the Orbitrap at 17,500 resolution. Precursor ion charge state screening was enabled and all unassigned charge states, as well as singly charged species, were rejected.

TMT-labelled peptides were analysed using a Dionex RSLCnano HPLC-coupled Tribrid Fusion mass spectrometer (Thermo Fisher Scientific). Peptides were first loaded onto a 2 cm PepMap trap column (100 μm) in 2% acetonitrile +0.1% formic acid. Trapped peptides were then separated on an analytical column (75 μm x 50 cm PepMap-C18 column) using the following mobile phases: 2% acetonitrile +0.1% formic acid (Solvent A) and 80% acetonitrile +0.1% formic acid (Solvent B). The linear gradient began with 5% B to 35% B over 220 min with a constant flow rate of 200 nl/min. The peptide eluent flowed into a nanoelectrospray emitter at the front end of either a Q-Exactive, or Q-Exactive Plus (quadrupole Orbitrap) mass spectrometer (Thermo Fisher Scientific). A max cycle time acquisition method was used (2 s). The primary mass spectrometry scan (MS1) was performed in the Oribtrap at 120,000 resolution. Then, the top N most abundant m/z signals were chosen from the primary scan for CID (30%) and Rapid-mode MS2 analysis in the linear ion trap. A synchronous precursor selection (SPS) method was employed. MS2 product ions were selected using four notches with a maximum injection time of 300 ms, fragmented by HCD (55%), and TMT tags mass analysed in the Orbitrap at 60,000 resolution. Precursor ion charge state screening was enabled and all unassigned charge states, as well as singly charged species, were rejected.

## MS data analysis

The SILAC and TMT RAW data files produced by the mass spectrometer were analysed using the quantitative proteomics software MaxQuant, versions 1.5.1.2 and 1.5.3.30 (*Cox and Mann, 2008*). This version of MaxQuant includes an integrated search engine, Andromeda (*Cox et al., 2011*). The database supplied to the search engine for peptide identifications was a UniProt human protein

database ('Human Reference Proteome' retrieved on April 16, 2016) combined with a commonly observed contaminants list. The initial mass tolerance was set to 7 p.p.m. and MS/MS mass tolerance was 20 ppm. The digestion enzyme was set to trypsin/P with up to two missed cleavages. Deamidation, oxidation of methionine and Gln->pyro Glu were searched as variable modifications. Identification was set to a false discovery rate of 1%. To achieve reliable identifications, all proteins were accepted based on the criteria that the number of forward hits in the database was at least 100-fold higher than the number of reverse database hits, thus resulting in a false discovery rate of less than 1%. Protein isoforms and proteins that cannot be distinguished based on the peptides identified are grouped by MaxQuant and displayed on a single line with multiple UniProt identifiers. The label-free quantitation (LFQ) algorithm in MaxQuant was used for protein quantitation. The algorithm has been previously described (Cox 2014). MaxQuant output was analysed in RStudio v1.0.136 with R v3.3.0. Protein quantitation was performed on unmodified peptides and peptides that have modifications that are known to occur during sample processing (pyro-Glu, deamidation). All resulting MS data were integrated and managed using PepTracker Data Manager, a laboratory information management system (LIMS) that is part of the PepTracker software platform (http://www.PepTracker.com).

For the interphase proteomic dataset, proteins were filtered to eliminate potential contaminants and reverse hits. For each of the four biological replicates, normalised SILAC ratios were normalised to the ratio measured in G1. A median ratio was calculated between replicates. The maximum fold change across a single experiment (i.e. between G1, S, G2, and M phase) was calculated. The distribution of maximum fold changes was then used to calculate fold change cutoffs were using Z-scoring, that is the median ±1.96 * the sample standard deviation. p-Values were calculated using a one-way ANOVA with the four biological replicates, excluding proteins with missing values. The p-value cutoff was arbitrarily set to 0.05. Proteins that met the fold change and p-value cutoffs were deemed significantly different in abundance between the cell cycle fractions.

SILAC ratios in the mitotic subphases dataset were normalised to the prophase fraction of the first replicate. The TMT reporter channels, 126, 127N, 128C, 129N, 130C, and 131, relate to G2, M, Pro, PM2, PM1, and Ana fractions, respectively. TMT reporter intensities were corrected for isotope contamination (Lot QL226165A) and normalised by dividing each TMT reporter by the sum of TMT reporter intensities. TMT reporter ratios were then calculated by dividing by the prophase (Pro) TMT reporter intensity. Thus TMT and SILAC experiments are both normalised to the same fraction. The TMT and SILAC datasets were merged using the Leading Razor Protein UniProt identifier. Correlation coefficients between the three biological replicates were calculated for the patterns across Pro, PM1, PM2, and Ana fractions in a pairwise fashion ('cor 1 vs 2', 'cor 1 vs 3', and 'cor 2 vs 3' in *Supplementary file 4*). A fold change cutoff based on maximum fold change was calculated using Z-scoring as above. Proteins that a) have at least one pair of biological replicates with a correlation coefficient greater than 0, and b) meet the fold change cutoff were deemed to be significantly changed.

Protein profiles were clustered using k-means, where k = 12. The appropriate k was determined by identifying the inflection point in a plot of within groups sum of squares vs the number of clusters. Clusters showing increased abundance were removed and remaining clusters were agglomerated into three groups based on 'earliest' fraction showing decreased abundance.

## Acknowledgements

This work was supported by funding from the Wellcome Trust (AIL, 105024/Z/14/Z, 108058/Z/15/Z), the EU EpiGeneSys network (AIL, HEALTH-F4-2010-257082) and the Knut and Alice Wallenberg Foundation (EL). We thank the UK Research Partnership Investment Fund and the Scottish Funding Council (Project H13047) for proteomics instrumentation funding and the Wellcome Trust (097418/Z/11/Z) for supporting the Flow Cytometry and Cell Sorting Facility at the University of Dundee. We thank Barbara Mann and Cassandra Pelletier for assisting with initial experiments on the TPX2 mutants. Long term time-lapse imaging was performed in the Nikon Center of Excellence in the IALS core facility at UMass Amherst. We thank Dr. J Chambers for expert assistance with microscopy and Matt Adler for assistance with quantification of mitotic duration. We thank our colleagues in the Lamond group for advice and discussion. We thank Calum Thompson and Alan Prescott (Dundee

Light Microscopy Facility), the Swedlow laboratory, and Raffaella Pippa (Lamond lab) for technical advice and assistance.

## Additional information

### Funding

| Funder | Grant reference number | Author |
|---|---|---|
| Knut och Alice Wallenbergs Stiftelse | | Emma Lundberg |
| Wellcome | 105024/Z/14/Z | Angus I Lamond |
| Wellcome | 108058/Z/15/Z | Angus I Lamond |
| European Commission | HEALTH-F4-2010-257082 | Angus I Lamond |
| Higher Education Funding Council for England | UK Research Partnership Investment Fund, Project H13047 | Angus I Lamond |
| Scottish Funding Council | Project H13047 | Angus I Lamond |

The funders had no role in study design, data collection and interpretation, or the decision to submit the work for publication.

### Author contributions

Tony Ly, Conceptualization, Data curation, Formal analysis, Supervision, Investigation, Methodology, Writing—original draft, Writing—review and editing; Arlene Whigham, Rosemary Clarke, Methodology; Alejandro J Brenes-Murillo, Visualization; Brett Estes, Formal analysis, Investigation; Diana Madhessian, Emma Lundberg, Formal analysis, Validation, Investigation, Writing—review and editing; Patricia Wadsworth, Conceptualization, Formal analysis, Funding acquisition, Validation, Methodology, Writing—original draft, Writing—review and editing; Angus I Lamond, Conceptualization, Formal analysis, Supervision, Funding acquisition, Methodology, Writing—original draft, Project administration, Writing—review and editing

### Author ORCIDs

Tony Ly https://orcid.org/0000-0002-8650-5215
Angus I Lamond http://orcid.org/0000-0001-6204-6045

### Decision letter and Author response

Decision letter https://doi.org/10.7554/eLife.27574.029
Author response https://doi.org/10.7554/eLife.27574.030

## Additional files

### Supplementary files

• Supplementary file 1. The effect of fixation and permeabilisation protocols on MS-based protein quantitation. A tab-delimited text file containing a list of all the protein groups identified and their associated SILAC ratios comparing the different fixation and permeabilisation protocols.
DOI: https://doi.org/10.7554/eLife.27574.023

• Supplementary file 2. Analysis of protein accumulation across interphase and mitosis. The table consists of an excel file containing two worksheets. The first worksheet lists the protein groups identified and their associated SILAC ratios in four biological replicates. For each biological replicate, the ratios were normalised to the ratio measured in G1. An offset was then added to the G1 ratio to account for the difference in time between cell division and an average G1 cell (calculated from *Figure 5B*). The second worksheet also lists the same protein groups, but with the unnormalised ratios. These are the ratios that were used to produce the 'neeps' plot in *Figure 5A*.
DOI: https://doi.org/10.7554/eLife.27574.024

• Supplementary file 3. Analysis of protein phosphorylation across interphase and mitosis. The table consists of a tab-delimited file containing the phosphorylation sites measured, quality measures (PEP, Score), and TMT ratios calculated relative to the G1 fraction from the two biological replicates. B – biological replicate, fc – fold change, repcor – Pearson's correlation score between the ratio patterns of the two biological replicates

DOI: https://doi.org/10.7554/eLife.27574.025

• Supplementary file 4. Analysis of protein abundances during mitotic subphases. The table consists of a tab-delimited file containing the proteins identified, quality measures (Q-value, Score, number of peptides), TMT ratios calculated relative to the prophase fraction, and SILAC ratios calculated relative to the prophase fraction in biological duplicate. cor – Pearson's correlation score between the ratio patterns of the three biological replicates (only mitotic subphases are compared). numcor – number of times the Pearson's correlation score is greater than 0.

DOI: https://doi.org/10.7554/eLife.27574.026

• Transparent reporting form

DOI: https://doi.org/10.7554/eLife.27574.027

## Major datasets

The following dataset was generated:

| Author(s) | Year | Dataset title | Dataset URL | Database, license, and accessibility information |
|---|---|---|---|---|
| Ly T, Whigham A, Clarke R, Brenes-Murillo A, Estes B, Wadsworth P, Lamond AI | 2017 | Proteomic analysis of cell cycle progression in asynchronous cultures, including mitotic subphases, using PRIMMUS | https://www.ebi.ac.uk/pride/archive/projects/PXD007787 | Publicly available at EBI PRIDE (accession no. PXD007787) |

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
