## [Decision Letter]

Thank you for submitting your article "Proteomic analysis of cell cycle progression in asynchronous cultures, including mitotic subphases, using PRIMMUS" for consideration by *eLife*. Your article has been favorably evaluated by Tony Hunter (Senior Editor) and three reviewers, one of whom is a member of our Board of Reviewing Editors.

The reviewers have discussed the reviews with one another and the Reviewing Editor has drafted this decision to help you prepare a revised submission. Our general recommendation is to convert this to a Tools and Resources paper to emphasize the novelty and utility of the technique (points 1-4 below), with less of an emphasis on the novel conclusions (points 5 and 6). Please let us know if you approve of this change from a Research Article.

Summary:

In this study the authors have developed an approach to combine fluorescence activated cell sorting with mass spectrometry analysis. This has enabled them to increase the temporal resolution of their proteomic analysis of the cell cycle to distinguish between different phases of mitosis. The authors use this new method (PRIMMUS) to analyse both protein abundance and phosphorylation through mitosis, taking ribonucleotide reductase subunit 2 and TPX2 as their respective examples. This is a very thorough study and the development of this technique will not only be of interest to those studying the cell cycle because it has the potential to analyse protein changes in any set of cells that can be distinguished using a fluorescent marker. The manuscript should, however, be strengthened by including a more thorough description of the experimental methods and analyses used. Where the manuscript is weaker is in the analysis of the TPX2 and RRM2 proteins. There are a number of controls missing and the data do not substantially extend our knowledge of spindle assembly and the means by which TPX2 phosphorylation perturbs Eg5 function is not properly followed up. The RRM2 data nicely show that degradation requires a neddylation event but does not go far enough to explain the discrepancy with the D'Angiolella study. In summary, this paper stands as an interesting technical development and resource but the TPX2 data should be reduced to make the paper more focused and concise.

Essential revisions:

1) A proper description of the methods is lacking in some sections. Please improve the description of the FACS method, the sources of antibodies, the cell numbers used in experiments and protein concentrations in lysates. In addition, the phosphopeptide analysis is a substantial part of the paper, yet there is no description of how the data analysis was carried out. The authors should include a section outlining phosphoproteomics mass spectrometry data analysis (site assignment software used, confidence cut-off, etc.), and how centrosome separation at nuclear envelope breakdown was measured.

2) Please include a description of the statistical methods used, and the biological and technical variability, for each analysis.

3) The authors highlight that cross-linking will lead to unusual modifications, compromising the protein identifications – have they used open searches to explore these effects on protein identification? Can the effects be quantified and then used to evaluate the impact of this on the quantitation? Was a filter for the number of peptides to be cut off included? If so, how is this controlled in the phosphorylation analysis which will be based on single peptides, since the effect of cross-linking modifications was only evaluated on proteins not on phosphorylation?

4) Was the protein phosphorylation analysis corrected for protein abundance? If so how? Other studies have shown there is a high correlation between abundance and phosphorylation. Moreover, were comparisons based on equal number of cells or total protein loaded for the analyses? Were there any differences in the bulk protein levels in the mitotic sub-phases as observed for the other phases? If so, how were these normalised?

5) The experiments concerning TPX2 phosphorylation do not greatly advance our knowledge of spindle assembly, and while identifying important sites of phosphorylation they do not show that early riser G2 phosphorylation of TPX2 is in fact important. These analyses weaken the manuscript and should be condensed.

6) The analysis of RRM2 degradation is also rather superficial and although the results using a neddylation inhibitor are a good first step they are not sufficiently strong to resolve the discrepancy with the published literature. These results should also be de-emphasised.

---

## [Author Response]

Essential revisions:1) A proper description of the methods is lacking in some sections. Please improve the description of the FACS method, the sources of antibodies, the cell numbers used in experiments and protein concentrations in lysates. In addition, the phosphopeptide analysis is a substantial part of the paper, yet there is no description of how the data analysis was carried out. The authors should include a section outlining phosphoproteomics mass spectrometry data analysis (site assignment software used, confidence cut-off, etc.), and how centrosome separation at nuclear envelope breakdown was measured.

We welcome the opportunity to expand the details of the methods used in our study. We therefore now include a detailed description of the FACS method, sources of the antibodies including RRID identifiers, and cell numbers used in the experiments. Where available, we also provide lysate protein concentrations. The Materials and methods section has been significantly expanded overall and now includes a section on how the phosphopeptide data were analysed. The centrosome separation data have been removed.

2) Please include a description of the statistical methods used, and the biological and technical variability, for each analysis.

A description of the statistical methods along with all of the available biological/technical variability data have now been included (Results, Figure captions, and Materials and methods).

3) The authors highlight that cross-linking will lead to unusual modifications, compromising the protein identifications – have they used open searches to explore these effects on protein identification? Can the effects be quantified and then used to evaluate the impact of this on the quantitation?

We thank the reviewer for highlighting the issue of modifications that have been shown to occur on amino acid residues due to formaldehyde fixation. We performed a data-dependent peptide search using MaxQuant and the raw files associated with the interphase mass spectrometry data. The data-dependent peptide search is a type of secondary ‘open search’ that identifies mass changes to unmodified peptides identified in a first pass search. Mass changes that would be consistent with previously shown formaldehyde-induced modifications contribute to 0.08% of the peptides identified. We now include text describing these results in the revised manuscript and include a table with the% of the methylene and methoyl adducts detected. We feel that this is a very useful addition to the overall study and are grateful for it being suggested in the review process.

Was a filter for the number of peptides to be cut off included?

We agree that filtering steps must be included to control identification confidence. While number of peptides historically has been used for this purpose, in common with the current consensus thinking in the field, we prefer to rely on the FDR approach. This not only controls for identification confidence but provides a statistical estimate of false discovery. Here we have implemented a stringent value of 1% FDR for protein identifications.

If so, how is this controlled in the phosphorylation analysis which will be based on single peptides, since the effect of cross-linking modifications was only evaluated on proteins not on phosphorylation?

To address this point, we performed an additional experiment, comparing the number of phosphorylation sites identified between control lysates and lysates from fixed, permeabilised cells in technical triplicate, i.e. three separate aliquots of lysate, processed on three different days. These data show that the relative number of phosphorylation sites identified are not significantly different between the two treatment conditions. Therefore, we conclude that the fixation and permeabilisation protocol does not significantly reduce the number of phosphorylation sites detectable. These data have now been included and discussed in the relevant sections of the revised manuscript.

4) Was the protein phosphorylation analysis corrected for protein abundance? If so how? Other studies have shown there is a high correlation between abundance and phosphorylation.

The supplementary tables include both phosphorylation ratios and ratios normalised to protein abundance. Figure 5 shows phosphorylation ratios only. We now include a supplementary figure (Figure 5—figure supplement 1) where the clustering analysis is repeated with ratios normalised to protein abundance using measurements from ‘total protein’ samples that were not phosphoenriched.

Moreover, were comparisons based on equal number of cells or total protein loaded for the analyses?

The mixing was based on equal number of cells, using FACS to accurately measure the numbers of cells used.

Were there any differences in the bulk protein levels in the mitotic sub-phases as observed for the other phases? If so, how were these normalised?

A review of the historical literature using radioactive nuclide labelled amino acids and synchronised cells to measure protein, RNA, and DNA content in mitotic cells revealed no reports of significant difference in bulk protein levels. Therefore, no significant difference in total protein content between mitotic sub-phases was anticipated in NB4 cells. Consistent with this idea, we find that less than 2% of proteins quantitated in our experiment show significant decreases between the four mitotic sub-phases analysed, as discussed in the revised manuscript.

5) The experiments concerning TPX2 phosphorylation do not greatly advance our knowledge of spindle assembly, and while identifying important sites of phosphorylation they do not show that early riser G2 phosphorylation of TPX2 is in fact important. These analyses weaken the manuscript and should be condensed.

As a general response to these comments from the review, we respectfully point out that the focus of our study was not a targeted analysis of spindle assembly. Rather, we present a significant new technical advance that has enabled a pioneering analysis at a systems level of protein abundance and phosphorylation changes across the cell cycle in unperturbed, asynchronously growing cells. In performing global analyses using new techniques, it is always valuable to select examples of novel findings for validation and deeper analysis. Our follow-on analysis here of the functional importance of phosphorylation of TPX2 at serine 738 was one such example of orthogonal validation of the MS data and exploration of the functional importance of the changes we observed. As such, we feel these data on TPX2 add significant value and biological interest to our findings and we are confident that most readers of our manuscript would share this view.

However, to accommodate the reviewers’ request, in this revised version we have now condensed the section of the manuscript that discusses phosphorylation of TPX2 at serine 738. Specifically, we removed the data using overexpression of mutant protein and also removed the data on the cold stability of spindle fibers. We replaced the data for the knock-down – rescue experiment with a knock-down – rescue experiment in cells expressing TPX2, and the mutant forms, from a bacterial artificial chromosome (BAC) and added time-lapse imaging using the BAC cell lines. These data have now been condensed into a single figure in the revised manuscript (Figure 7).

The results we show in Figure 7 provide clear validation that this cell cycle-varying TPX2 phosphorylation site that we identified by MS analysis is indeed functionally important for mitosis in vivo. We acknowledge that our data do not address the separate issue of how the timing of the phosphorylation event impacts cell division. As explained above, this was not the focus of our study. A better understanding of the function of TPX2 prior to NEBD is needed to develop assays for early phenotypes and while we agree strongly that such experiments will be interesting, they are clearly beyond the scope of this present manuscript.

In summary, we think that these comprehensive changes to the manuscript and new data that we have added in response to the reviewers’ comments convey the functional importance of this phosphorylation site in TPX2 that was detected by our global MS analysis without detracting from the main focus of our study.

6) The analysis of RRM2 degradation is also rather superficial and although the results using a neddylation inhibitor are a good first step they are not sufficiently strong to resolve the discrepancy with the published literature. These results should also be de-emphasised.

The potential discrepancy with the published literature is the timing of RRM2 degradation. RRM2 was previously reported to be degraded in G2 phase. Using MS-based proteomics, we showed that RRM2 levels remaining high in G2, but are decreased by prometaphase. Independently, using an antibody against RRM2, we showed by flow cytometry that RRM2 levels decrease in prometaphase, consistent with the MS results.

In the revision, we include several new data figures testing the antibody’s specificity against RRM2. Flow cytometry signal from the RRM2 antibody decreased significantly in cells depleted of RRM2 using siRNA. Moreover, immunofluorescence from the anti-RRM2 antibody shows correlated localisation with GFP signal in cells overexpressing of RRM2-GFP. These data together support that the anti-RRM2 antibody is specific for RRM2.

Using this validated antibody, our collaborators at the Human Protein Atlas independently evaluated RRM2 abundance during the cell cycle in U2OS cells. RRM2 levels were highest in G2 as shown by immunostaining of RRM2 in FUCCI-expressing U2OS cells. Co-immunostaining of cyclin B1 and RRM2 levels show that RRM2 levels decrease after cyclin B1 signal is observed in the nucleus, i.e. either at, or shortly after, entry into mitosis. These data support our conclusion from the MS analysis that RRM2 levels are high during G2 and decrease during early mitosis.

Collectively, all of our data, using different cell lines and generated in different laboratories, strongly argue that RRM2 levels are high during G2 and do not substantially decrease until early mitosis. Importantly, these experiments were performed in minimally or unperturbed cells without synchronisation, using an antibody that was validated for specificity. We note that the purported discrepancy stems from a study in the literature where antibody validation, the antibody source, and details about how the antibody was generated, were notably absent. We leave open the possibility that RRM2 degradation timing could differ in other cell types or cellular contexts (perhaps stress induced by synchronisation, for example). However, from our own data from two laboratories using multiple, independent techniques of measuring RRM2 levels in two human cell lines, we must conclude that RRM2 protein abundance is cell cycle regulated, peaks during G2, and its degradation does not substantially occur until early mitosis. We therefore feel these new findings will be of significant interest to the cell biology community.